 

# KymoButler, a deep learning software for automated kymograph analysis

Maximilian AH Jakobs*, Andrea Dimitracopoulos, Kristian Franze*

Department of Physiology, Development and Neuroscience, University of Cambridge, Cambridge, United Kingdom

**Abstract** Kymographs are graphical representations of spatial position over time, which are often used in biology to visualise the motion of fluorescent particles, molecules, vesicles, or organelles moving along a predictable path. Although in kymographs tracks of individual particles are qualitatively easily distinguished, their automated quantitative analysis is much more challenging. Kymographs often exhibit low signal-to-noise-ratios (SNRs), and available tools that automate their analysis usually require manual supervision. Here we developed KymoButler, a Deep Learning-based software to automatically track dynamic processes in kymographs. We demonstrate that KymoButler performs as well as expert manual data analysis on kymographs with complex particle trajectories from a variety of different biological systems. The software was packaged in a web-based 'one-click' application for use by the wider scientific community (https://deepmirror.ai/kymobutler). Our approach significantly speeds up data analysis, avoids unconscious bias, and represents another step towards the widespread adaptation of Machine Learning techniques in biological data analysis.

*For correspondence:
mj455@cam.ac.uk (MAHJ);
kf284@cam.ac.uk (KF)

Competing interest: See
page 16

Reviewing editor: Patricia
Bassereau, Institut Curie, France

## Introduction

Many processes in living cells are highly dynamic, and molecules, vesicles, and organelles diffuse or are transported along complex trajectories. Particle tracking algorithms represent powerful approaches to track the dynamics of such particles (*Jaqaman et al., 2008*; *Sbalzarini and Koumoutsakos, 2005*; *Lee and Park, 2018*). However, in many scenarios particles follow a distinct pathway within cells and move much faster than the confounding cell. For example, molecules transported along neuronal axons, dendrites, or along cilia typically move along the structure's long axis and do not show significant motion perpendicular to that path. Similarly, retrograde actin flow typically occurs along a single axis within the cell. Hence, when the cell is not moving significantly for the duration of imaging, one can define, for example manually draw, a so-called 'stationary path' (*Figure 1A*) along which particles move either forwards or backwards. In these cases, kymographs provide an elegant solution to the visualisation and analysis of particle dynamics.

To generate a kymograph, the intensity profile along the manually drawn stationary path (black dashed line in *Figure 1A*) is extracted for each frame of a time-lapse movie, and then these profiles are stacked into individual rows of an image (*Figure 1A*). In the resulting space-time image, each (usually fluorescently) labelled particle is shown as a line, whose slope, for example, represents the velocity of that particle (*Figure 1A*).

In many biological processes, multiple particles move along the same stationary path with little to no deviations, making kymographs a very useful representation of their dynamics. Hence, kymographs have been widely employed to visualise biological processes across different length scales, ranging from diffusion and transport of single molecules to whole cell movements (*Twelvetrees et al., 2016*; *Barry et al., 2015*). The analysis of these kymographs only requires tracing lines in 2D images, a rather simple task compared to the more general approach of particle

**eLife digest** Many molecules and structures within cells have to move about to do their job. Studying these movements is important to understand many biological processes, including the development of the brain or the spread of viruses.

Kymographs are images that represent the movement of particles in time and space. Unfortunately, tracing the lines that represent movement in kymographs of biological particles is hard to do automatically, so currently this analysis is done by hand. Manually annotating kymographs is tedious, time-consuming and prone to the researcher's unconscious bias.

In an effort to simplify the analysis of kymographs, Jakobs et al. have developed KymoButler, a software tool that can do it automatically. KymoButler uses artificial intelligence to trace the lines in a kymograph and extract the information about particle movement. It speeds up analysis of kymographs by between 50 and 250 times, and comparisons show that it is as reliable as manual analysis. KymoButler is also significantly more effective than any previously existing automatic kymograph analysis programme. To make KymoButler accessible, Jakobs et al. have also created a website with a drag-and-drop facility that allows researchers to easily use the tool.

KymoButler has been tested in many areas of biological research, from quantifying the movement of molecules in neurons to analysing the dynamics of the scaffolds that help cells keep their shape. This variety of applications showcases KymoButler's versatility, and its potential applications. Jakobs et al. are further contributing to the field of machine learning in biology with 'deepmirror.ai', an online hub with the goal of accelerating the adoption of artificial intelligence in biology.

---

tracking, where one has to identify the centre of the particles in each frame, and then correctly assign these coordinates to corresponding particles across frames.

Publicly available kymograph analysis software simplifies the tedious and time-consuming task of tracing kymographs, but most of these solutions require manual supervision and/or high signal to noise ratios (*Neumann et al., 2017*; *Mangeol et al., 2016*; *Chenouard, 2010*; *Zala et al., 2013*). These tools perform reasonably well when applied to particles with unidirectional motion and uniform velocities as, for example, growing microtubule +ends (*Figure 1C*, example 2) and F-actin dynamics in retrograde actin flow (*Lazarus et al., 2013*; *del Castillo et al., 2015*; *Alexandrova et al., 2008*; *Babich et al., 2012*).

In many other biological contexts, however, particles can stop moving, change velocity, change direction, merge, cross each other's path, or disappear for a few frames. The kymographs obtained from these processes exhibit 'bidirectional' motion (*Figure 1C*, example 1); this category includes cellular transport processes, for example molecular or vesicle transport in neuronal axons and dendrites (*Faits et al., 2016*; *Tanenbaum et al., 2013*; *Koseki et al., 2017*). Thus, the problem of automatically and reliably tracking dynamic processes in kymographs still leaves substantial room for improvement, and given the limitations of currently available kymograph analysis software, most kymographs are still analysed by hand, which is slow and prone to unconscious bias.

In recent years, Machine Learning (ML), and particularly Deep Neural Networks, have been very successfully introduced to data processing in biology and medicine (*Mathis, 2018*; *Weigert, 2017*; *Florian, 2017*; *Guerrero-Pena, 2018*; *Falk et al., 2019*; *Bates, 2017*). ML-based image analysis has several advantages over other approaches: it is less susceptible to bias than manual annotation, it takes a much shorter time to analyse large datasets, and, most importantly, it comes closer to human performance than conventional algorithms (*Mathis, 2018*).

Most ML approaches to image analysis utilise **F**ully **C**onvolutional Deep Neural **N**etworks (FCNs) that were shown to excel at object detection in images (*Dai, 2016*; *Szegedy, 2014*; *LeCun et al., 1989*; *Falk et al., 2019*). Through several rounds of optimisation, FCNs select the best possible operations by exploiting a multitude of hidden layers. These layers apply image convolutions using kernels of different shapes and sizes, aiming to best match the output of the neural network to the provided training data labels, which were previously derived from manual annotation. This means that the network learns to interpret the images based on the available data, and not on a priori considerations. This approach has become possible due to the dramatic improvements in computation times of modern CPUs and the adoption of GPUs that can execute an enormous number of

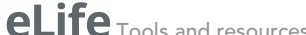

**Figure 1.** Kymograph generation and KymoButler.

The online version of this article includes the following figure supplement(s) for figure 1:

**Figure supplement 1.** Example kymographs and software workflow.

**Figure supplement 2.** The software modules in detail.

**Figure supplement 3.** Synthetic training data examples.

**Figure supplement 4.** Geometric mean of track recall and precision for different trackness thresholds.

**Figure supplement 5.** Geometric mean of track recall and precision for different signal to noise ratios and particle densities.

operations in parallel. Currently, the most successful architecture for biological and medical image analysis is the U-Net, which takes an input image to generate a binary map that highlights objects of interest based on the training data (*Ronneberger et al., 2015*).

Here we present KymoButler, a new stand-alone FCN software based on the U-Net architecture, to automatically and reliably extract particle tracks from kymographs. The software is packaged into an easy-to-use web interface (https://deepmirror.ai/kymobutler) and a downloadable software package, and it was benchmarked against traditional software and manual annotation on synthetic (i.e., ground truth) and real (biological) data. We show that KymoButler performs as well as manual annotation on challenging bidirectional kymographs, where particles disappear, reappear, merge, cross each other's path, move in any direction, change speed, immobilise, and reverse direction. KymoButler thus represents a substantial improvement in the automation of kymograph tracing, speeding up the experimental workflow, while preserving the accuracy of manual annotations.

## Results

### The KymoButler software package

For our FCN-based kymograph analysis software, we implemented a customised architecture based on U-Net (*Ronneberger et al., 2015*). This architecture comprises two segmentation networks ('modules'), one specialised on kymographs with exclusively unidirectional particle movements, the other one on bi-directional kymographs. These segmentation networks were trained to binarize the image into regions with particle tracks (foreground) and noise (background). They take an input kymograph to generate 2D maps that assign a 'trackness' value between 0 and 1 to each pixel of the input image, with higher values representing a higher likelihood of pixels being part of a track (see Materials and methods).

Our training (95%) and validation (5%) data consisted of manually annotated tracks in 487 unidirectional and 79 bidirectional kymographs (unpublished data from our group and other laboratories, see Materials and methods and Acknowledgements for details). Since no ground truth was available in the manually annotated kymographs, we also generated 221 synthetic unidirectional and 21 synthetic bidirectional kymographs that were used for training.

The unidirectional segmentation module generates separate trackness maps for tracks with negative and positive slopes (which could, for example, correspond to tracks of anterograde and retrograde transport processes, respectively), to remove line crossings from the output (see Materials and methods). Since particles have uniform speeds, individual tracks can be extracted via binarization of the trackness map.

In bidirectional kymographs, tracks show more complex morphologies, since they can change direction/speed and cross each other multiple times. The bidirectional segmentation module therefore generates a single trackness map, which needs to be further processed in order to obtain individual particle tracks. In particular, one has to resolve crossings between tracks. We did this by implementing a decision module which iterates through all crossings to find the most likely final segmentation (see Materials and methods for details).

We found the binarization thresholds for both modules to depend on the biological application and on the signal to noise ratio of the input image. However, we observed the best performance for both segmentation modules generally for values between 0.1–0.3 (*Figure 1—figure supplement 4*).

Finally, our software has to decide whether to analyse a given kymograph with the unidirectional module or the bidirectional module. Therefore, we implemented a 'classification module' that

classifies input kymographs into unidirectional or bidirectional ones. We linked the class module to the unidirectional and bidirectional segmentation modules as well as to the decision module and packaged them into KymoButler, an easy-to-use, drag and drop browser-based app for quick and fully automated analysis of individual kymographs (https://deepmirror.ai/kymobutler).

The only free parameter in KymoButler is the threshold for trackness map segmentation. The default threshold is set to 0.2, but users can freely adjust it for their specific application. After the computation, which takes 1–20 s per kymograph (depending on complexity), KymoButler generates several files including a dilated overlay image highlighting all the tracks found in different colours, a CSV file containing all track coordinates, a summary file with post processing data, such as average velocities and directionality, and preliminary plots of these quantities (*Figure 1B*). KymoButler worked well on previously published kymographs from a variety of different biological data (*Figure 1C* and *Figure 1—figure supplement 1A*) and on unpublished data from collaborators (*Figure 2—figure supplement 1B* and *Figure 3—figure supplement 2*).

## Performance on unidirectional kymographs

We quantitatively evaluated the performance of KymoButler on unidirectional kymographs, that is particles that move with mostly uniform velocities and with no change in direction (*Figure 1C*, *Figure 2*, *Figure 1—figure supplement 1A*). The unidirectional module of KymoButler was compared to an existing kymograph analysis software, which is based on Fourier filters, and which provided the best performance among publicly available software in our hands (KymographDirect package; *Mangeol et al., 2016*). Additionally, we traced kymographs by hand to obtain a control for the software packages.

First, we generated 10 synthetic movies depicting unidirectional particle dynamics with low signal-to-noise ratio (~1.2, see Materials and methods) and extracted kymographs from those movies using the KymographClear (*Mangeol et al., 2016*) Fiji plugin. Each of the kymographs was then analysed by Fourier-filtering (KymographDirect), KymoButler, and by hand, and the identified trajectories overlaid with the ground truth (i.e., the known dynamics of the simulated data) (*Figure 2A*). KymoButler typically took less than a minute to analyse the 10 kymographs while fourier filtering took about 10 min since thresholds had to be set individually for each image. Manual annotation by an expert took about 1.5 hr.

To quantify the quality of the predicted traces, we first determined the best predicted track for each ground truth track (in case several segments were predicted to cover the same track) and then calculated the fraction of the length of the ground truth track that was correctly identified by that predicted track ('track recall') (*Figure 2B*). Additionally, we determined the best overlapping ground truth track for each predicted track and then calculated the fraction of the length of the predicted track that was overlapping with the ground truth track ('track precision'). Examples of low/high precision and low/high recall are shown in *Figure 2B*. We then calculated the geometric mean of the average track recall and the average track precision (the 'track F1 score', see Materials and methods) for each kymograph (*Figure 2E*). The median F1 score of the manual control was 0.90, KymoButler achieved 0.93, while Fourier filtering achieved a significantly lower F1 score of 0.63 ($p = 4 \cdot 10^{-5}$, Kruskal-Wallis Test, Tukey post-hoc: manual vs KymoButler $p = 0.6$, manual vs Fourier Filtering $p = 3 \cdot 10^{-3}$).

Our synthetic data intentionally included gaps of exponentially distributed lengths (see Materials and methods), allowing us to quantify the ability of KymoButler to bridge gaps in kymograph tracks (*Figure 2C*, F), which are frequently encountered in kymographs extracted from fluorescence data (*Applegate et al., 2011*). Both KymoButler and manual annotation consistently bridged gaps that belonged to the same trajectory, while Fourier filtering was less accurate (89% of all gaps correctly bridged by KymoButler, 88% by manual, and 72% by Fourier filter analysis; median of all 10 synthetic kymographs, $p = 10^{-4}$, Kruskal-Wallis Test, Tukey post-hoc: manual vs KymoButler $p = 0.9$, manual vs Fourier Filtering $p = 2 \cdot 10^{-3}$, Figure 2F).

We also quantified the ability of KymoButler to resolve track crossings (*Figure 2D*). Again, both KymoButler and manual annotation performed significantly better than Fourier filtering (88% KymoButler, 86% manual, 60% Fourier filter; median percentage of correctly resolved crossings of all 10 synthetic kymographs, $p = 10^{-4}$, Kruskal-Wallis Test, Tukey post-hoc: manual vs KymoButler $p = 0.9$, manual vs Fourier Filtering $p = 1 \cdot 10^{-3}$, *Figure 2G*).

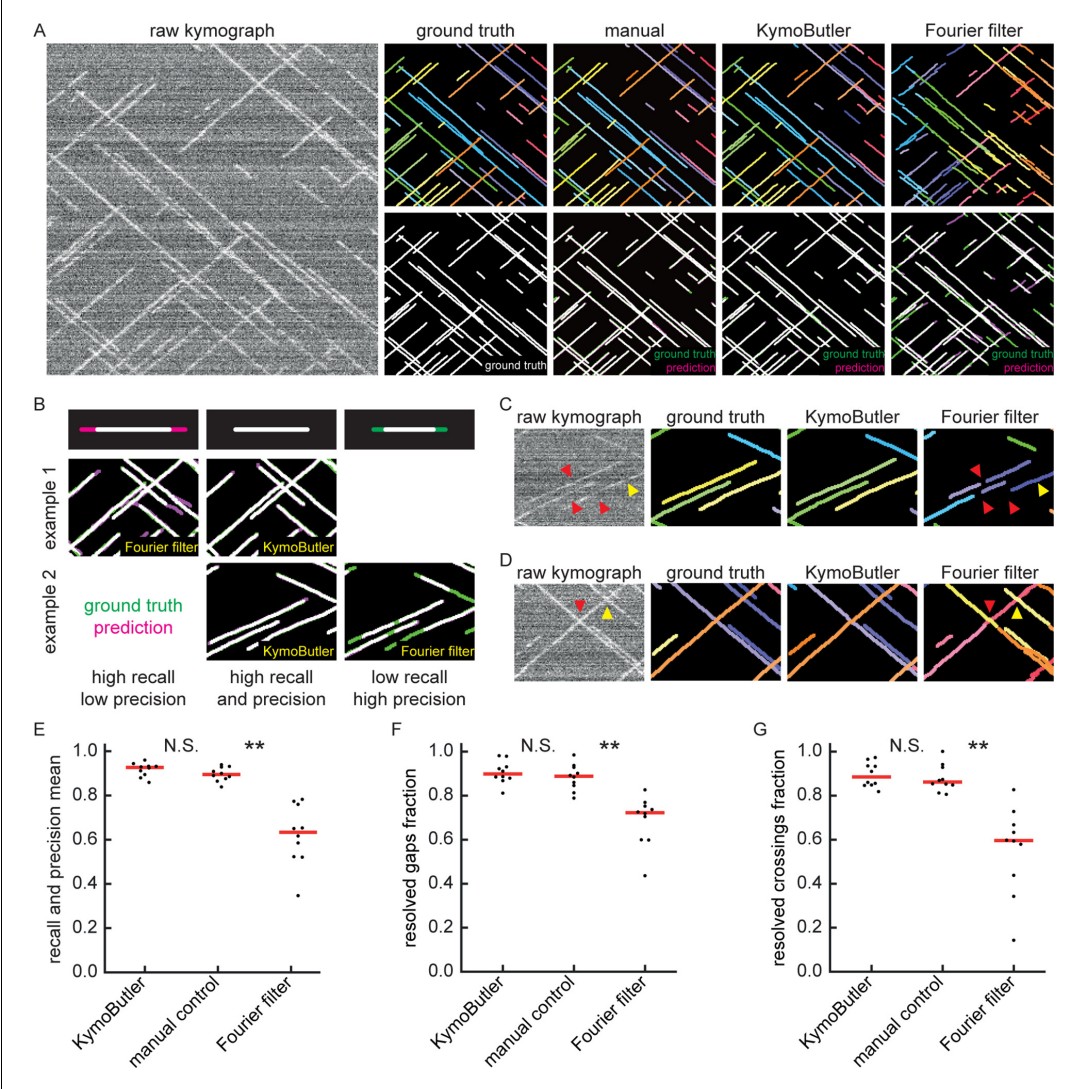

**Figure 2.** Benchmark of KymoButler against unidirectional synthetic data. (**A**) An example synthetic kymograph and its corresponding ground truth, manual control, the prediction by KymoButler, and the prediction by Fourier filtering. The top row depicts individual tracks in different colours and the bottom row shows the prediction overlay (magenta) with the ground truth (green) for all approaches. Discrepancies are thus highlighted in magenta (false positive) and green (false negative), while matching ground truth and prediction appears white. (**B**) Schematic explaining the concept of recall and precision. The top row depicts the possible deviations of the prediction from the ground truth. The middle and bottom rows show example overlays, again in green and magenta, from the synthetic data. In the left column, the prediction is larger than the ground truth (magenta is visible) leading to false positive pixels and low track precision, but a small number of false negatives and thus high track recall. An example prediction overlay of the Fourier filter approach is shown, which tends to elongate track ends. The right column shows a shorter prediction than the ground truth, leading to green segments in the overlay. While this prediction has high track precision (low number of false positive pixels), track recall is low due to the large number of false negatives. Again, a cut-out from the Fourier filter prediction is shown, where multiple gaps are introduced in tracks, thus severely diminishing track recall (see Material and methods for a detailed explanation of recall and precision). The middle column shows the same two cut outs analysed by KymoButler. No magenta or green segments are visible, thus leading to high recall and precision. (**C**) Synthetic kymograph region with four gaps highlighted (arrow heads): in one or more kymograph image rows the signal was artificially eliminated but kept in the ground truth to simulate real fluorescence data. While KymoButler efficiently connects tracks over gaps, the Fourier filter is unable to do so and breaks up those tracks into segments or incorrectly shortens these tracks (red arrow heads). Yellow arrow heads depict correct gap bridging events. (**D**) A synthetic kymograph with several line crossings. While KymoButler efficiently resolved all crossings, that is lines that cross other lines are not broken up into two segments, the Fourier filter correctly identifies the line crossing at the yellow arrow head but erroneously terminates the red and yellow tracks at the red arrow head. (**E**) The geometric means of recall and precision ('track F1 score') for KymoButler, the Fourier filter approach, and manual control. Each dot represents the average track F1 score of one synthetic kymograph ($p = 4 \cdot 10^{-5}$, Kruskal-Wallis Test, Tukey post-hoc: manual vs KymoButler $p = 0.6$, manual vs Fourier Filtering $p = 3 \cdot 10^{-3}$). (**F**) Quantification of gap bridging performance for KymoButler (89%), manual control (88%), and Fourier filter (72%); lines: medians of all 10 synthetic kymographs, $p = 10^{-4}$, Kruskal-Wallis Test, Tukey post-hoc: manual vs KymoButler $p = 0.9$, manual vs Fourier

*Figure 2 continued on next page*

*Figure 2 continued*

Filtering $p = 2 \cdot 10^{-3}$. (**G**) The fraction of correctly identified crossings for KymoButler, manual annotation, and the Fourier filter (88% KymoButler, 86% manual, 60% Fourier filter; lines: medians of all 10 synthetic kymographs, $p = 10^{-4}$, Kruskal-Wallis Test, Tukey post-hoc: manual vs KymoButler $p = 0.9$, manual vs Fourier Filtering $p = 1 \cdot 10^{-3}$). Tracks smaller than 3 pixels and shorter than 3 frames were discarded from the quantification.

The online version of this article includes the following source data and figure supplement(s) for figure 2:

**Source data 1.** Table of presented data.
**Source data 2.** Synthetic kymographs and movies.
**Figure supplement 1.** Data quantities derived from unidirectional kymographs using manual annotation, KymoButler, and Fourier filtering for simulated and real data.

Remarkably, KymoButler was able to correctly pick up ~80% of all tracks at an SNR as low as 1.1, where tracks are barely visible by eye (*Figure 1—figure supplement 5B,D*), and at high particle densities (~70% of the kymograph image covered with signal) (*Figure 1—figure supplement 5F,H*). Manual annotation at such extremes performed similarly to KymoButler (*Figure 1—figure supplement 5C,D,G,H*).

Finally, we compared KymoButler's overall performance on kymographs containing unidirectional traces with that of alternative analysis approaches. KymoButler performed similar to or better than manual annotation of synthetic data when analysing particle velocities, directionality, travel time, travel distance, and particle numbers, while the Fourier filter frequently deviated by more than 50% from ground truth averaged values (*Figure 2—figure supplement 1A*). When testing KymoButler's overall performance on real kymographs of our validation data set, we compared deviations from manual annotation as no ground truth was available. KymoButler deviated by less than 10% from most manual estimates (but found ~30% more particles), while the Fourier filter approach deviated by up to 50% from the manually calculated values (*Figure 2—figure supplement 1B*).

In summary, KymoButler was able to reliably track particle traces in kymographs at low SNR and high particle densities in both synthetic and real data, and it clearly outperformed currently existing automated software, while being as consistent as manual expert analysis while being ~100 x faster.

## KymoButler's performance on bidirectional Kymographs

As in many kymographs obtained from biological samples trajectories are not unidirectional, we also tested the performance of KymoButler on complex bidirectional kymographs, that is of particles with wildly different sizes, velocities, and fluorescence intensities that frequently change direction, may become stationary and then resume motion again (see *Figure 1B,C*, *Figure 3A*, *Figure 1—figure supplement 1A* for examples). Available fully automated software that relied on edge detection performed very poorly on our synthetic kymographs (*Figure 3—figure supplement 1*). Therefore, we implemented a custom-written wavelet coefficient filtering algorithm to compare our FCN-based approach to a more traditional non-ML approach (*Figure 3A*, *Figure 3—figure supplement 1*, Materials and methods). In short, the wavelet filtering algorithm generates a trackness map, similar to KymoButler, by applying a stationary wavelet transform to the kymograph to generate so-called 'coefficient images' that highlight horizontal or vertical lines. These coefficient images are then overlaid and binarized with a fixed value (0.3), skeletonised, and fed into the KymoButler algorithm without the decision module, that is crossings are resolved by linear regression prediction.

We generated 10 kymographs from our synthetic movies with the KymographClear package (average signal-to-noise ratio was 1.4, since any lower signal generally obscured very faint and fast tracks). Each of the kymographs was then analysed by wavelet coefficient filtering, KymoButler, and manual annotation, and the predicted traces overlaid with the ground truth (*Figure 3A*). While the wavelet approach and KymoButler were able to analyse the 10 kymographs in less than 1 min, manual annotation by an expert took about 1.5 hr. Moreover, whereas the manual annotation and KymoButler segmentation overlaid well with the ground truth, the wavelet approach yielded numerous small but important deviations.

Similarly to the unidirectional case, we quantified track precision and recall (*Figure 3B, E*) and calculated the resolved gap fraction (*Figure 3C, F*) and crossing fraction (*Figure 3D, G*). The median of the track F1 scores per kymograph for manual annotation (0.82) was similar to KymoButler (0.78), while the wavelet filter approach only gave 0.61 ($p = 7 \cdot 10^{-5}$, Kruskal-Wallis Test, Tukey post-hoc:

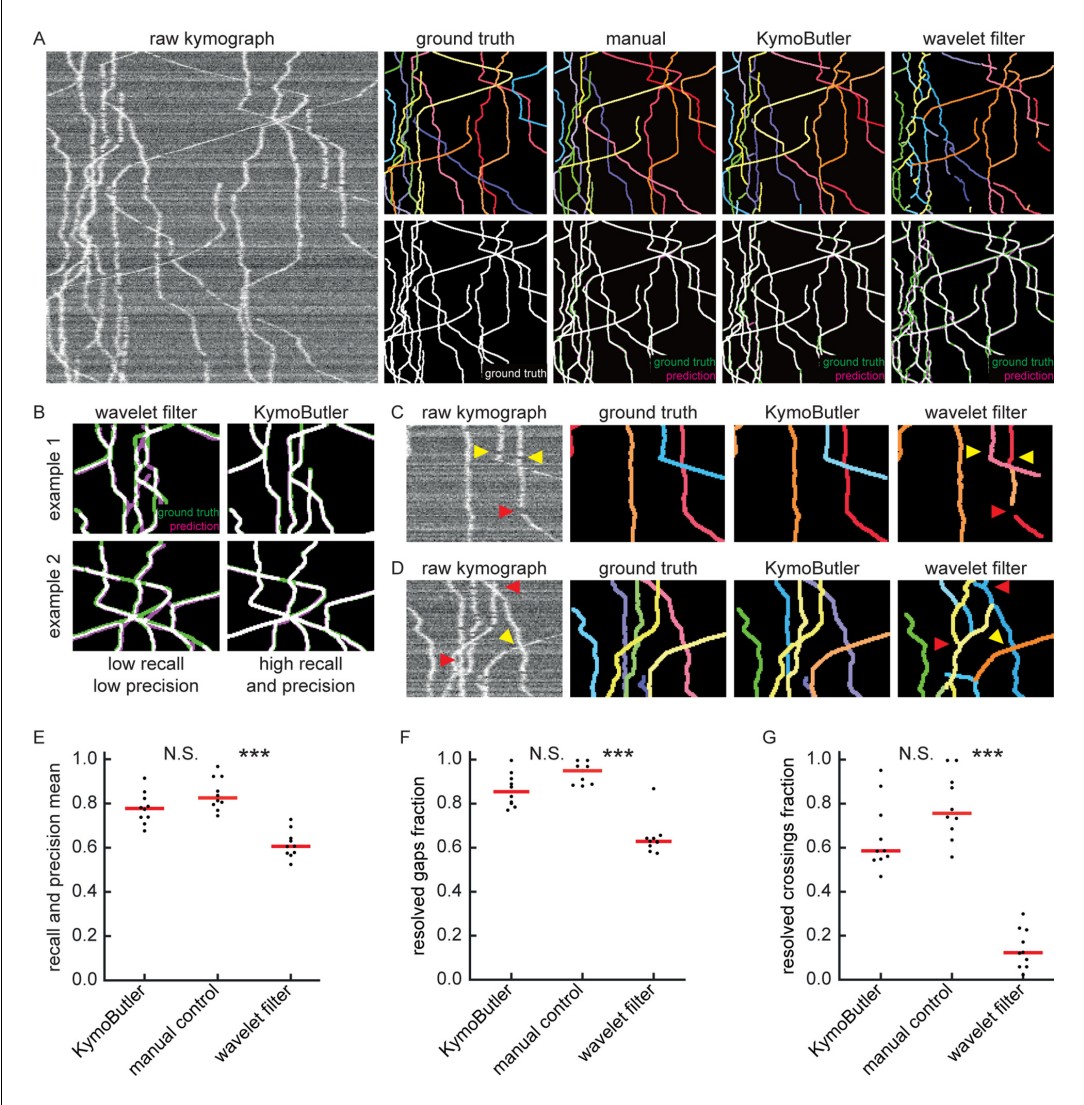

**Figure 3.** Benchmark of KymoButler against complex bidirectional synthetic data. (**A**) Example synthetic kymograph and its corresponding ground truth, manual control, the prediction by KymoButler, and the prediction via wavelet coefficient filtering. The top row depicts individual tracks in different colours and the bottom row shows the prediction overlay (magenta) with the ground truth (green) for all approaches. Discrepancies are highlighted in magenta (false positive) and green (false negative), while the match of ground truth and prediction appears white. (**B**) Example recall and precision of KymoButler and wavelet filtering. While KymoButler shows high recall and high precision, the wavelet filter approach yields significant deviations from the ground truth (green and magenta pixels). (**C**) Synthetic kymograph region with three artificial gaps highlighted (arrow heads). While KymoButler efficiently connects tracks over gaps, the wavelet filter is unable to do so and breaks up those tracks into segments (red arrow heads). The yellow arrow heads depict correct gap bridging events. (**D**) A synthetic kymograph with several line crossings. While KymoButler efficiently resolved all crossings, that is lines that cross other lines are not broken up into segments, the wavelet filter only resolves one crossing correctly (yellow arrow head). (**E**) The geometric means of track recall and track precision (track F1 score) for KymoButler, manual control, and the wavelet filter. Each dot represents the average F1 score of one synthetic kymograph ($p = 8 \cdot 10^{-5}$, Kruskal-Wallis Test, Tukey post-hoc: manual vs KymoButler $p = 0.7$, manual vs wavelet filtering $p = 10^{-4}$). (**F**) Quantification of gap performance for KymoButler, manual annotation, and wavelet filter ($p = 3 \cdot 10^{-4}$, Kruskal-Wallis Test, Tukey post-hoc: manual vs KymoButler $p = 0.4$, manual vs wavelet filtering $p = 2 \cdot 10^{-4}$). (**G**) The fraction of resolved crossings for KymoButler, manual control, and the wavelet filter ($p = 3 \cdot 10^{-5}$, Kruskal-Wallis Test, Tukey post-hoc: manual vs KymoButler $p = 0.4$, manual vs wavelet filtering $p = 2 \cdot 10^{-5}$). KymoButler identifies tracks in complex kymographs as precisely as manual annotation by an expert.

The online version of this article includes the following source data and figure supplement(s) for figure 3:

**Source data 1.** Table of presented data.
**Source data 2.** Synthetic kymographs and movies.
**Figure supplement 1.** Performance of different skeletisation techniques on a synthetic bidirectional kymograph.
**Figure supplement 2.** Synthetic data quantities derived from bidirectional kymographs using manual annotation, KymoButler, and wavelet filtering for simulated and real data.

manual vs KymoButler $p = 0.3$, manual vs wavelet filtering $p = 10^{-4}$, *Figure 3E*). While gaps were resolved by KymoButler and manual annotation in 86% and 95% of cases, respectively, only 63% were resolved by the wavelet algorithm (median of all 10 synthetic kymographs, $p = 4 \cdot 10^{-5}$, Kruskal-Wallis Test, Tukey post-hoc: manual vs KymoButler $p = 0.2$, manual vs wavelet filtering $p = 10^{-5}$, *Figure 3F*). Crossings were rarely resolved correctly by the wavelet algorithm (13%) but much more reliably by KymoButler (59%) and manual annotation (76%) (median of all 10 synthetic kymographs, $p = 3 \cdot 10^{-5}$, Kruskal-Wallis Test, Tukey post-hoc: manual vs KymoButler $p = 0.4$, manual vs wavelet filtering $p = 4 \cdot 10^{-3}$, *Figure 3G*).

As bidirectional tracks varied in intensity, lower SNRs obscured faint tacks so that performance dropped slightly faster with decreasing SNR than for unidirectional tracks (*Figure 1—figure supplement 5A,D*), and it decreased linearly for increasing particle densities (*Figure 1—figure supplement 5E,H*). Manual annotation at high signal densities and low SNRs again yielded similar results as KymoButler (*Figure 1—figure supplement 5C,D,G,H*). The analysis of quantity averages of 10 synthetic kymographs revealed that KymoButler was as accurate as manual annotation, while the wavelet filter deviated significantly more from the ground truth (*Figure 3—figure supplement 2A*). We also tested KymoButler on our validation dataset. Results were similar to the performance on synthetic data (~10% deviation from manual annotation), and the wavelet filter performed significantly worse than KymoButler (*Figure 3—figure supplement 2B*).

Overall, these results showed that KymoButler performs well on both unidirectional and bidirectional kymographs, clearly outperforms currently available automated analysis of kymographs, and it performs as well as manual tracing, while being much faster and not prone to unconscious bias.

## Discussion

In this work, we developed software based on Deep Learning techniques to automate the tracking of dynamic particles along a stationary path in a noisy cellular environment. Convolutional neural networks (CNNs) are nowadays widely applied for image recognition. Since tracking is a priori a visual problem, we built a modular software utilising CNNs for identifying tracks in kymographs. We deployed our networks as KymoButler, a software package that takes kymographs as inputs and outputs all tracks found in the image in a matter of seconds. The network outperforms standard image filtering techniques on synthetic data as well as on kymographs from a wide range of biological processes, while being as precise as expert manual annotation.

The KymoButler software has only one adjustable detection parameter that is left to the user: a sensitivity threshold that, if low, allows more ambiguous tracks to be recognised, and if high discards them. For our synthetic data, the best value for the threshold lay between 0.1 and 0.3 (*Figure 1—figure supplement 4*), and we observed a similar range for a variety of kymographs from published data. However, the threshold depends on the SNR of the input images, so that the correct threshold has to be chosen based on each biological application and imaging conditions. Furthermore, the performance of both KymoButler and manual annotation decreased with decreasing SNR and increasing particle density (number of crossings in the image, *Figure 1—figure supplement 4*). Note that the particle density here also depends on the particle's frequency of change in direction in dense kymographs as more bidirectional particles tend to cover larger proportions of the kymograph image. Hence, we strongly recommend to visually inspect the output of KymoButler for each new application, and to compare the output to manual annotation.

Most of the publicly available kymograph analysis software requires manual labelling to extract quantitative data (*Chenouard, 2010*; *Neumann et al., 2017*; *Zala et al., 2013*). Some automated approaches have been published in the context of specific biological questions, but since these programs are currently not publicly available it is not clear how well they would perform on kymographs from other applications (*Mukherjee et al., 2011*; *Reis et al., 2012*). Other approaches do not extract individual tracks but only macroscopic quantities, as for example velocities (*Chan and Odde, 2008*). As KymoButler is fully automated and able to reliably analyze kymographs from a wide range of biological applications, it fills an important gap. Here we showed that KymoButler is able to quantify mitochondria movement in neuronal dendrites, microtubule growth dynamics in axons, and in vitro dynamics of single cytoplasmic dynein proteins (*Figure 1* and *Figure 1—figure supplement 1*). The training and validation data for KymoButler comprised kymographs depicting the dynamics of microtubule +ends, mitochondria movement, molecular motor movements, and vesicle transport in

neuronal processes (example kymographs: *Lazarus et al., 2013*; *Cioni et al., 2019*; *Hangen et al., 2018*; *Gerson-Gurwitz et al., 2011*). Hence, KymoButler will perform best on similar data. However, we predict that it can furthermore be applied to most other kymographs obtained from time-lapse fluorescence microscopy without the need of any modifications.

KymoButler outperformed Fourier filtering, edge detection, and customised wavelet coefficient selection on synthetic kymographs. While Fourier filtering 'only' performed ~30% worse than Kymo-Butler on synthetic unidirectional kymographs, edge detection on synthetic bidirectional kymographs suffered greatly from background fluctuations and low SNR to such an extent that the extracted data was unusable (see *Figure 3—figure supplement 1* for one example). Therefore, we designed a filtering algorithm based on wavelet coefficient image selection to analyse complex bidirectional kymographs specifically for our synthetic data. KymoButler still performed 20% better than this approach (*Figure 3*). The main problem with either filtering approach compared to KymoButler was their inability to bridge track gaps and resolve line crossings, both of which occur frequently in biological data (*Figures 2C,D* and *3C,D*). These challenges were met by KymoButler, which performed as well as expert annotation, but within a much shorter time (*Figures 2* and *3*). Consequently, KymoButler generated similar measures of averaged quantities (average velocities, displacements, etc.) as manual annotation (*Figure 2—figure supplement 1A* and *Figure 3—figure supplement 2A*).

Synthetic kymographs, however, will reproduce the complexity of real kymographs only to some degree, as they exhibit homogenous background, no artefacts, no varying particle intensity in time, and individual tracks can appear rather similar. Hence, we also benchmarked KymoButler for both the unidirectional and bidirectional (real) validation datasets. KymoButler calculated similar average quantities as obtained from manual annotation, such as average particle velocities and pausing times, with some minor deviations (*Figure 2—figure supplement 1B* and *Figure 3—figure supplement 2B*). However, since values obtained from manual annotation deviated by up to 20% from ground truth values on synthetic data, deviations from manual annotation should not automatically be interpreted as an erroneous deviation (*Figure 2—figure supplement 1* and *Figure 3—figure supplement 2*).

Our results show that KymoButler is able to correctly identify individual full-length tracks in kymographs with an average track F1 score (geometric mean of track precision and recall) of 92% on unidirectional tracks and 78% on complex bidirectional tracks (similar to manual annotation), without suffering from inconsistency, bias, and laborious tracing, that plague manual tracking. While Kymo-Butler is already performing very well, we aim to significantly improve it over future iterations. Every time someone uses our webform, s/he has the option to anonymously upload the kymograph to our cloud. Once a large number of diverse kymographs is uploaded, these kymographs will be annotated by us and added to our training data, improving KymoButler even further.

The ultimate challenge will be to expand our approach to 2D or even 3D tracking problems. Here, we defined a 1D region of interest in 2D time-lapse movies, extracted 2D (space and time) images (kymographs), and finally tracked 2D lines in those images. A similar, albeit computationally heavier, approach could stack the frames of a 2D/3D movie on top of each other to generate a 3D/4D kymogram (2D space and time, or 3D space and time). Previously generated kymograms have led to intriguing results on whole-cell particle tracking problems with high SNR (*Racine et al., 2007*). The use of higher dimensional FCNs in the future has great potential to yield human-like performance on any biological and medical tracking problems.

## Materials and methods

**Key resources table**

| Resource | Designation. | Source. | Identifiers. | Additional information |
|---|---|---|---|---|
| Software, algorithm | MATLAB | MATLAB | RRID:SCR_001622 | Used for statistical analysis |

*Continued on next page*

*Continued*

| Resource | Designation. | Source. | Identifiers. | Additional information |
|---|---|---|---|---|
| Software, algorithm | Fiji | Fiji is Just ImageJ (https://fiji.sc) | RRID:SCR_002285 | Used to generate and analyse kymographs with KymographClear/ Direct https://sites.google. com/site/kymographanalysis/ |
| Software, algorithm | Wolfram Mathematica | Wolfram Mathematica | RRID:SCR_014448 | Code available under https://github.com/MaxJakobs/KymoButler (copy archived at swh:1:rev:e35173e9051eb5395f9b13dcd8f487ffa4098592) |

All code was written in the wolfram language in Mathematica https://wolfram.com/mathematica and, if not stated otherwise, can be found online under: https://github.com/MaxJakobs/KymoButler (copy archived at swh:1:rev:e35173e9051eb5395f9b13dcd8f487ffa4098592).

## The KymoButler software package

The KymoButler software was implemented in Mathematica to take advantage of easy web form deployment and distribution. The workflow is shown in *Figure 1—figure supplement 1B*. Our approach was to first segment kymograph pixels that are part of particle tracks from pixels that were part of the background with our segmentation modules. From previous work we knew that kymographs that depict unidirectional movement only, can be filtered into tracks that have positive slope and those that have negative slope (*Chenouard, 2010*), while no such assumptions can be made about bidirectional kymographs. Hence, we decided to take advantage of this simplification of unidirectional kymograph analysis by training two modules: one that is specialised to segment unidirectional kymographs and another one that segments bidirectional ones. Note that the bidirectional module is able to analyze any kymograph, including unidirectional ones, but since it is not specialised it performs slightly worse than the unidirectional module on unidirectional kymographs. To further simplify software usability, we prepended a class module that classifies input kymographs as bidirectional or unidirectional, and then applies the corresponding segmentation module and, if bidirectional, decision module. Our downloadable software package on GitHub allows the user to call either segmentation module (unidirectional/bidirectional) directly, if they wish to do so.

When the kymograph is classified as unidirectional by the class module, the unidirectional segmentation module generates two trackness score maps for particles with negative or positive slope (*Figure 1—figure supplement 1B*). Since the particles move with roughly the same velocity, the resulting maps mostly do not exhibit any crossings. Thus, we binarize the maps with a threshold between 0.1–0.3 (see benchmarking section for more information about the threshold). The resulting binary maps are then thinned iteratively so that each trace is only one pixel wide at any point and pruned so that branches that are shorter than three pixels are deleted. Subsequently, each trace is segmented and selected only if they are at least three frames long and three pixels large (these values can be varied by the user if needed but were kept constant throughout this manuscript for unidirectional kymographs). In the final step, pixels that lie in the same row of the kymograph are averaged over so that the final track has only one entry per frame.

For bidirectional kymographs the software generates a trackness map, applies a binarization threshold (0.1-0.3, see benchmarking for more details), iterative thinning, and pruning (minimum length 3 pixels). Similar to the unidirectional case, our software allows the selection of tracks with a minimum number of pixels and/or frames. However, since the resulting skeletonised map had a substantial number of crossings and could not be easily segmented to yield individual tracks, we implemented a further module in the software. First, all lines in the skeletonised map are shortened so that each white pixel at a track end only has neighbouring pixels in different rows (time dimension). This was done so that we could detect track starting points ('seeds') with a Hit-Miss transformation with kernel: $\begin{pmatrix} -1 & -1 & -1 \\ -1 & 1 & -1 \\ 0 & 0 & 0 \end{pmatrix}$. Application of this kernel yielded a binary map with 0 everywhere except at track seeds (*Figure 1—figure supplement 1B*, red dots). These seeds were then used to start tracing individual tracks in the kymograph by always advancing to the next white pixel. Once more than one potential future pixel is encountered, the decision module is called. The module

generates three 48x48 crops of (1) the input kymograph, (2) the skeletonised trackness map, and (3) the skeleton of the current track and predicts a trackness map that has high values on the skeleton segment of the most likely future track (*Figure 1—figure supplement 1B*). This map is binarized with threshold 0.5 and thinned. The precise threshold had little effect on the final output, so we fixed it at 0.5 for all applications. Users can vary this threshold as well in the source code on GitHub. Next, the largest connected component in the map is selected as the most likely future path and appended to the track if longer than 2 pixels. The average trackness value of this component (from the decision module prediction) is saved as a measure of decision 'confidence'. This process is repeated until no further possible pixels are found or no future path is predicted which is when the track is terminated. Once all seeds are terminated, the software subtracts all the found paths from the skeletonised trackness map and again looks for new seeds which are then again tracked in the full skeletonised image. The process is repeated until no further seeds are found, and then all tracks are averaged over their timepoints (rows in the kymograph image). Subsequently the software deletes tracks that are shorter than 5 pixels or part of another track and assigns overlaps that are longer than 10 pixels to the track with the highest average decision confidence.

Both the unidirectional and the bidirectional module output a coloured overlay in which each track is drawn in a different randomly assigned colour and dilated with factor one for better visibility (see *Figure 1B–C* and *Figure 1—figure supplement 1A*). Additionally, the software generates one CSV file that contains all the track coordinates, a summary CSV file that gives derived quantities, such as track direction and average speed, and plots depicting these quantities.

The software was deployed from Mathematica as a cloud based interface (https://deepmirror.ai/kymobutler) and a Mathematica package (https://github.com/MaxJakobs/KymoButler; copy archived at swh:1:rev:e35173e9051eb5395f9b13dcd8f487ffa4098592).

## Network architectures

Our networks were built from convBlocks (a convolutional layer with $3 \times 3$ kernel size, padding, and arbitrary number of output channels followed by a batch normalisation layer and a 'leaky' ramp (leakyReLU) activation function ($leakyReLU(x): = max(x, 0) - 0.1\ max(-x, 0)$). Batch normalisation is useful to stabilise the training procedure as it rescales the inputs of the activation function (leakyReLu), so that they have zero mean and unit variance. The leakyReLu prevents the so-called 'dead ReLu's' by applying a small gradient to values below 0. These building blocks were previously used for image recognition tasks in *Google's* inception architecture and in the U-Net architecture (*Szegedy, 2014*; *Falk et al., 2019*).

The module architectures we settled on are shown in *Figure 1—figure supplements 1–2*. All modules used the same core building blocks while having different input and output ports. The classification module takes a resized kymograph of size $64 \times 64$ pixels and generates two output values that correspond to the class probabilities for unidirectional/bidirectional kymographs (*Figure 1—figure supplement 2A*). The unidirectional segmentation module takes one input kymograph and generates two output images that correspond to the trackness scores of particles with positive or negative slopes (*Figure 1—figure supplement 2B*). The bidirectional segmentation module takes one input kymograph and generates one trackness score map highlighting any found particle tracks (*Figure 1—figure supplement 2C*). Finally, the decision module takes three inputs of size $48 \times 48$ pixels to generate one trackness map (*Figure 1—figure supplement 2D*). All modules share the same core network that is essentially a U-Net with padded convolutions and with 64 (in the top level) to 1024 (in the lowest level) feature maps. We experimented with more complex architectures (parallel convolution modules instead of blocks, different number of feature maps) but could only observe minor increase in accuracy at a large expense in computation time. Due to the U-Net architecture, each dimension of the inputs to the segmentation modules needs to be a multiple of 16. Thus, inputs were resized when they did not match the dimension requirements, and then the binarized output images from the segmentation modules were resized to the original input image size before proceeding further.

## Network training

To train the networks we quantified the difference between their output $o$ and the desired target output $t$ through a cross entropy loss layer ($CEloss(t, o) = -(t \cdot ln(o) + (1 - t) \cdot ln(1 - o))$). The loss was

averaged over all output entries (pixels and classes) of each network. While we tried other loss functions, specifically weighted cross entropy loss and neighbour dependent loss as described in *Bates (2017)*, we persistently obtained higher track precision and track recall with the basic cross entropy loss above.

Our training data comprised a mixture of synthetic data and manually annotated unpublished kymographs, kindly provided by the research groups mentioned in the acknowledgements. Most of the manual annotation was done by M. A. H. J. and A. D. In total, we used 487 (+200 synthetic) unidirectional, and 79 (+21 synthetic) bidirectional kymographs, with 95% of the data used for network training, and ~5% of retained for network validation. All network training was performed on a workstation, using a nVidia 1080 Ti or a nVidia 1070 GPU.

The class module depicted in *Figure 1—figure supplement 2A* was trained with batches of size 50 (with 25 unidirectional and 25 bidirectional kymographs to counter class imbalance) with random image transformations that included image reflections, rotations, resizing, colour negation, gaussian noise, random noise, and random background gradients. The final input image was randomly cropped to 64 × 64 pixels (see examples *Figure 1—figure supplement 3A*) and the class module was trained using stochastic gradient descent (ADAM optimiser, *Kingma, 2017*, initial learning rate 0.001), until the validation set error rate was consistently 0%.

The unidirectional segmentation module (*Figure 1—figure supplement 2B*) was trained with batches comprising 20 randomly selected kymographs from our training set (example in *Figure 1—figure supplement 3B*). We applied the following image transformations: Random reflections along either axis, random 180-degree rotations, random cropping to 128 × 80 pixels (approximately the size of our smallest kymograph), random gaussian and uniform noise, and random background gradients. Note that we did not apply any resizing to the raw kymograph since that generally decreased net performance. Additionally, we added Dropout Layers (10–20%) along the contracting path of our custom U-Net to improve regularisation. Each kymograph in this training set was generated by hand with KymographTracker (*Chenouard, 2010*), but to increase dataset variability we took the line profiles from KymographTracker and generated kymographs with a custom Mathematica script that applied wavelet filtering to the plotted profiles. The resulting kymographs have a slightly different appearance than the ones created with KymographTracker and are thus useful to regularise our training process. Several modules were trained until convergence and the best performing one (according to the validation score) was selected (ADAM optimiser, initial learning rate of 0.001, learning rate schedule = $If[batch<4000, 1, .5]$).

The bidirectional segmentation module (*Figure 1—figure supplement 2C*, example data *Figure 1—figure supplement 3C*) was trained in the same way as the unidirectional segmentation module, with the exception of a slightly different learning rate schedule ($If[batch<3000, 1, .5]$). Additionally, since we did not have access to many of the original movies from which the kymographs were generated, we could not generate kymographs with different algorithms as done for the unidirectional module.

Training data for the decision module (*Figure 1—figure supplement 2D*) was obtained from the bidirectional (synthetic +real) kymographs by first finding all the branch points in a given ground truth or manually annotated image. Then, each track was separated into multiple segments, that go from its start point to a branching point or its end point. For each branchpoint encountered while following a track, all segments that ended within 3 pixels of the branchpoint were selected. Then, (1) a 48 × 48 pixel crop of the raw kymograph around the branchpoint, (2) a binary map representing the track segment upstream of the branching point (centred with its end in pixel coordinates 25,25, with image padding applied if the end was close to an image corner), and (3) the corresponding 48 × 48 pixel region in the binary image representing all possible paths were used as inputs to the decision module. The binary image representing the ground truth or annotated future segment downstream of the branchpoint was used as the target image (see *Figure 1—figure supplement 3D* for an example training set). Thus, the training set comprised three input images and one output image which we used to train the decision module. To increase the module's focus on the non-binary raw kymograph crop, we applied 50% dropout to the full skeletonised input and 5% dropout to the input segment. As explained above, we used random image augmentation steps like reflections, rotations, gaussian +uniform noise. Additionally, we employed random morphological thinning to the binary input/output images to simulate artefacts. Several networks were trained until convergence (pixel

wise cross entropy loss, ADAM optimiser, initial learning rate 0.001, batch size 50, learning rate schedule $If[batch<8000, \ 1, \ .5])$, and the best performing one was selected.

## Synthetic data

Synthetic data was generated by simulating individual particles on a stationary path of length 300 pixels for 300 frames to generate 300 × 300 pixel kymographs. To obtain unidirectional particles we seeded 30 + 30 particles with negative or positive slope at random timepoints/positions. Next, a random velocity between 1–3 pixels/frame was chosen for all particles in the movie, with a random noise factor to allow slight changes in velocity, and a particle PSF between 3–6 pixels. Each particle was assigned a survival time drawn from an exponential distribution with scale 0.01, after which it would disappear. Gaps of random length (exponentially distributed) were subsequently assigned to each track individually. From these tracks we then generated a kymograph with gaussian noise, used for neural network training, and a 20 × 300 pixel movie with 300 frames for benchmarking. The resulting kymographs and movies had an average signal-to-noise ratio of 1.2 (calculated as the average intensity of the signal, divided by the average intensity of the background). Finally, we removed tracks that overlapped for the whole duration of their lifetime.

To obtain synthetic data of complex bidirectional particle movements, we generated datasets with either 15 tracks (for benchmarking) or 30 tracks (for training) per movie. The maximum velocity was set to three pixels/frame, as above this velocity it became hard to manually segment tracks from kymographs. Each movie was assigned a random velocity noise factor between 0 and 1.5 pixels/ frame, a random switching probability between 0 and 0.1 (to switch between stationary and directed movement) and a random velocity flipping factor between 0 and 0.1 (to flip the direction of the velocity). Individual particles were simulated by first calculating their lifetime from an exponential distribution with scale 0.001. Then, a random initial state, moving or stationary, was selected as well as a random initial velocity and a particle size between 1–6 pixel. In the simulation, particles could randomly switch between different modes of movement (stationary/directed), flip velocities and were constantly subjected random velocity noise (movie specific). Finally, tracks that were occulted by other tracks were removed, and a movie (used for benchmarking) and a kymograph (used for training) were generated. The resulting kymographs and movies had an average signal-to-noise ratio of 1.4.

## Benchmarking

In order to benchmark the performance of software and manual predictions, we implemented a custom track F1 score which was calculated as the geometric mean of track recall and track precision. To calculate track recall, each ground truth track was first compared to its corresponding predicted track, and the fractional overlap between them was calculated. Since predicted tracks do not necessarily follow the exact same route through a kymograph, but frequently show small deviations from the ground truth (see *Figure 3* and *Figure 3—figure supplement 1*) we allowed for a 3.2-pixel deviation from the ground truth (two diagonal pixels). The maximum fractional overlap was then selected and stored as the track recall. The recall was thus one when the full length of a ground truth track was predicted, and 0 if the track was not found in the prediction. We would like to highlight that this criterion is very strict: if a ground truth track is predicted to be two tracks (for example, by failing to bridge a gap along the track), the recall fraction would decrease by up to 50%, even if most of the pixels are segmented correctly and belong to predicted tracks.

Track precision was calculated by finding the largest ground truth track that corresponded, that is had the largest overlap, to each prediction, and then calculating the fraction of the predicted track that overlapped to the ground truth track. Therefore, a track precision of 1 corresponded to a predicted track that was fully part of a ground truth track while a precision of 0 meant that the predicted track was not found in the ground truth. In general, increasing precision leads to a lower recall and vice versa, so that taking the track F1 score as the geometric mean between the two is a good measure of overall prediction performance.

To quantify gap performance, we searched for track segments within 3 pixels of the gap for each frame, to allow for predictions that deviated slightly from the ground truth. Once each frame of the gap was assigned to a corresponding predicted segment, the gap was deemed resolved. If one or more frames of the gap had no overlapping segment to the prediction, the gap was labelled

unresolved. Our synthetic tracks had 954 gaps in the 10 kymographs of unidirectional data, and 840 gaps in the 10 kymographs of bidirectional data, and the largest gap size was six pixels. For each kymograph, we then calculated the fraction of gaps resolved.

To quantify KymoButler performance on crossings, we first generated binary images for each ground truth track and calculated overlaps with other ground truth tracks by multiplying those images with each other. The resulting images had white dots wherever two tracks crossed. Those dots were then dilated by a factor of 16 to generate circles and overlaid with the original single-track binary image to generate binary maps that contain segments of ground truth tracks that cross/merge with other tracks. Next, we generated dilated (factor 1) binary maps for each predicted track and multiplied them with each of those cross segments to obtain the largest overlapping track for each segment. We then visually inspected a few examples and determined that an overlap of 70% corresponds to a correctly resolved crossing and allowed for slight variations in predicted tracks when compared to ground truth. Finally, we calculated the fraction of crossings resolved per kymograph.

Derived quantities were calculated as follows. For average velocities, we first calculated the absolute frame to frame displacement and from there the average frame to frame velocity per track and the average frame to frame velocity per kymograph. The absolute displacement was calculated as a sum of all absolute frame to frame displacements and then averaged to yield a measure per kymograph. The travel time was calculated as the absolute time a particle was visible in a given kymograph and averaged for each kymograph. Directionality per particle was calculated as the sign of the end to end displacement for unidirectional kymographs. For bidirectional kymographs, we first calculated the directionality of up to five frame long segments, which was +1 when all displacements in that segment were positive, −1 when all displacements were negative, and 0 otherwise. The sign of the sum of all segment directionalities was taken as a measure for the bidirectional track directionality. The pause time for each bidirectional particle was calculated as the number of segments with 0 displacement and averaged per kymograph. Finally, the percentage of reversing tracks was calculated by dividing the number of tracks that exhibit segments in both directions by all tracks. In *Figure 2—figure supplement 1* and *Figure 3—figure supplement 2*, we only show relative deviations from the manual annotation because we cannot disclose any data from the real, unpublished kymographs obtained from collaborators.

All statistical analysis was carried out in MATLAB using either a Wilcoxon rank-sum test or a Kruskal Wallis test (http://mathworks.com).

## Module performance evaluation

To benchmark the unidirectional segmentation module of KymoButler, we generated 10 synthetic movies of the dynamics of particles that move with uniform speed and do not change direction as described in the section about synthetic data generation. We then imported these movies into ImageJ (http://imagej.nih.gov) via the Kymograph Clear package (*Mangeol et al., 2016*), drew a profile by hand and generated kymographs from them. These kymographs were then imported into the KymographDirect software package (also *Mangeol et al., 2016*), Fourier filtered and thresholded to extract individual particle tracks. This approach required manual selection of the threshold for each individual kymograph. We additionally traced the same kymographs by hand in ImageJ to compare software performance to expert analysis. To find a suitable range of binarization thresholds for our unidirectional segmentation module we calculated the track wise F1 score on the 10 kymographs for thresholds between 0.05 and 0.5 (*Figure 1—figure supplement 4*). We observed the highest scores between 0.1 and 0.3 for both our synthetic data and other unpublished kymographs and also deemed these thresholds best by visual inspection of predicted kymograph tracks. Hence, we chose 0.2 as the segmentation map threshold to benchmark our predictions at.

In order to benchmark the bidirectional segmentation module and the decision module we generated 10 synthetic movies of the dynamics of complex bidirectional particles. These movies were imported into ImageJ with the KymographClear package and kymographs extracted. We subsequently tried to use the edge detection option in KymographDirect to extract individual tracks but were unable to obtain meaningful tracks (*Figure 3—figure supplement 1*). We also tried other options in the package but could not get good results on our synthetic data without substantial manual labour for each kymograph, defeating the goal of a fully automated analysis. Therefore, we wrote a custom script to carry out automated bidirectional kymograph analysis. We experimented

with a few different approaches (for example fourier-filtering and customised edge detection) and settled on wavelet coefficient filtering as it gave the highest F1 score on our test dataset. This algorithm applied a stationary wavelet transformation with Haar Wavelets (Mathematica wavelet package) to each kymograph to decompose the image into different coefficient images that highlight different details (for example vertical or horizontal lines). We then selected only those coefficient images that recapitulated particle traces in our synthetic kymographs. These images are overlaid and thresholded with an optimised threshold to generate binary maps that can be iteratively thinned to obtain a skeletonized 'trackness' map similar to the outputs of our segmentation modules. This map was then traced with the same algorithm as in our decision module. However, while the Kymo-Butler decision module used a neural network to predict path crossings, the wavelet filtering algorithm performed simple linear prediction by taking the dilated (factor 1) binary segment of a track and rotating it by 180 degrees. Then the 'prediction' was multiplied with the skeletonized trackness map and the largest connected component selected as the future path. In contrast to the original decision module, this approach does not yield any information about decision 'confidence'. Thus, to resolve track overlaps at the end of the algorithm, we randomly assigned each overlap to one track and deleted them from the others. Note that the wavelet approach was heavily optimised on our synthetic kymographs and performed poorly on generic real kymographs. We also traced the same 10 kymographs by hand in ImageJ. To find a suitable range of binarization thresholds for our bidirectional segmentation module we calculated the track wise F1 score for thresholds between 0.05 and 0.5 (*Figure 1—figure supplement 4*) and observed the same optimal range as the unidirectional segmentation module (0.1–0.3) for both our synthetic data and other unpublished kymographs. Hence, we chose 0.2 as the threshold score to benchmark our predictions.

## Acknowledgements

We would like to thank Eva Pillai for scientific input, proofreading, and logo design, Hannes Harbrecht for fruitful discussions about neural networks; Hendrik Schuermann, Ishaan Kapoor, and Kishen Chahwala for help with kymograph tracing; and the Mathematica stack exchange community (https://mathematica.stackexchange.com) without whom this project would have taken several decades longer. Unpublished kymographs to train KymoButler were provided by Caroline Bonnet (neurocampus, University of Bordeaux), Dr. Jean-Michel Cioni (San Raffaele Hospital, Milan), Dr. Julie Qiaojin Lin (University of Cambridge), Prof. Leah Gheber and Dr. Himanshu Pandey (Ben-Gurion University of the Negev), Dr. Carsten Janke and Satish Bodakuntla (Insitut Curie, Paris), and Dr. Timothy O'Leary and Adriano Bellotti (University of Cambridge). Additionally, we would like to thank the Bordeaux Imaging Center, part of the national infrastructure France BioImaging (ANR-10INBS-04–0), for valuable feedback on our software. We would also like to thank eLife and PLOS whose open access policy enabled us to show a variety of kymographs in this manuscript. The authors acknowledge funding by the Wellcome Trust (Research Grant 109145/Z/15/Z to MAHJ), the Herchel Smith Foundation (Fellowship to AD), Isaac Newton Trust (Research Grant 17.24(p) to KF), UK BBSRC (Research Project Grant BB/N006402/1 to KF), and the ERC (Consolidator Award 772426 to KF).

## Additional information

### Competing interests

Maximilian AH Jakobs, Andrea Dimitracopoulos: We launched deepmirror.ai as a platform to promote the use of AI-based technologies for biological data analysis. We will be publishing tutorials and sample code to help people get started with developing their own machine learning software. We also intend to publish our work on KymoButler and future publications of our AI-based software on the website. All of this will be free of charge and available to all. Further in the future, we plan to also start offering paid professional services for customers that want to set up custom AI-based software for applications, in case they are not covered by our research. This software may or may not be made available on deepmirror.ai, depending on our clients' requests. The other author declares that no competing interests exist.

## Funding

| Funder | Grant reference number | Author |
| --- | --- | --- |
| Wellcome Trust | 109145/Z/15/Z | Maximilian AH Jakobs |
| Herchel Smith Foundation | | Andrea Dimitracopoulos |
| Isaac Newton Trust | 17.24(p) | Kristian Franze |
| Biotechnology and Biological Sciences Research Council | BB/N006402/1 | Kristian Franze |
| European Research Council | 772426 | Kristian Franze |

The funders had no role in study design, data collection and interpretation, or the decision to submit the work for publication.

## Author contributions

Maximilian AH Jakobs, Conceptualization, Data curation, Software, Formal analysis, Funding acquisition, Validation, Investigation, Visualization, Methodology, Writing—original draft, Writing—review and editing; Andrea Dimitracopoulos, Software, Formal analysis, Funding acquisition, Methodology, Writing—review and editing; Kristian Franze, Conceptualization, Supervision, Funding acquisition, Investigation, Writing—original draft, Project administration, Writing—review and editing

## Author ORCIDs

Maximilian AH Jakobs https://orcid.org/0000-0002-0879-7937
Kristian Franze https://orcid.org/0000-0002-8425-7297

## Decision letter and Author response

Decision letter https://doi.org/10.7554/eLife.42288.sa1
Author response https://doi.org/10.7554/eLife.42288.sa2

# Additional files

## Supplementary files

• Transparent reporting form

## Data availability

All code, if not stated otherwise, can be found online under: https://github.com/MaxJakobs/KymoButler (copy archived at https://archive.softwareheritage.org/swh:1:dir:f47b5bc2e657f0c1f85a6-c9e622fbd608dfdc7fd). Source data for Figure 2 and 3 were uploaded as CSV files (Figure 2-source data 1 and Figure 3-source data 1) and the underlying datasets as zip files (Figure 2-source data 2 and Figure 3-source data 2).

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
