## [Decision Letter]

[Editors’ note: this article was originally rejected after discussions between the reviewers, but the authors were invited to resubmit after an appeal against the decision.]

Thank you for submitting your work entitled "KymoButler: A deep learning software for automated kymograph analysis" for consideration by *eLife*. Your article has been reviewed by three peer reviewers, and the evaluation has been overseen by a Reviewing Editor and a Senior Editor. The following individuals involved in review of your submission have agreed to reveal their identity: Jeff Urbach (Reviewer #1) and Gaudenz Danuser (Reviewer #2).

This manuscript describes KymoButler, a machine-learning software that has been developed for the automated analysis of kymographs. There is a big need in this area since kymograph analysis (or related linear tracking) is notoriously difficult to perform and to automate. Although the reviewers consider that KymoButler could potentially provide an open solution to automated kymograph tracking to the community, they eventually agree after discussion that in its current state, KymoButler has not been tested on challenging enough problems and so far, does not represent a general and new solution to complicated tracking problems. In particular, the reviewers were concerned about the performance of KymoButler for tracking elements on much more diverse traces that those considered in the manuscript: for instance, in the general case of axonal mobility tracking where cargoes or components can move in a direction, change speed, immobilize, reverse direction, and do any of this several times in a single trace. We feel that in its current form, this tool does not outperform some other existing tracking methods. For these reasons, we think that your manuscript does not meet the criteria of innovation of *eLife* and cannot be accepted for publication in the journal.

*Reviewer #1:*

This manuscript describes a very valuable contribution to the field of quantitative biological imaging. It is well written and clearly motivated, and I have no substantial concerns. The results look solid, the approach is creative and appropriate, and it seems likely that the software will be widely utilized. As with all machine learning applications, it would be nice to have a deeper understanding of how the software is actually working, but that isn't a reason not to share the results in their current form.

*Reviewer #2:*

This manuscript describes the use of a convolutional neural net (CNN) for the analysis of kymographs. While kymography is still used by the live cell imaging community to measure subcellular dynamics, and thus some tools to automate and reduce bias would be helpful, this work does not advance the existing repertoire of image analysis for this purpose.

First, the proposed method is not a kymograph analyzer, as claimed, but it relies on a CNN to learn a streak filter (the authors should have visualized the initial layers the trained net, and this would have become abundantly clear, I assume). The output of the CNN here is not a set of trajectories in the kymograph representation – the entity that eventually can be used to quantify particle velocities and, perhaps, lifetimes – but it offers merely a pixel-by-pixel map of a score that defines how probable the pixel is to be located on a streak. To get from the 'streakness' score map to the individual trajectories, the authors have to threshold the map, and then apply a morphological operator to thin the high score mask to single-pixel chains. Anyone who has written a program for line or edge detection knows, these two steps are much harder and are the decisive ones for the final results. These are the steps where the algorithm needs to account for streak crossings, junctions, and gaps. The authors largely wipe this under the rug. The results in Figures 2 and 3 reveal numerous places where exactly these two steps went wrong. And the authors probably intentionally pick data examples with relatively low particle density and/or fairly uniform particle motion. As the particle density and motion heterogeneity between and within trajectories increases, these limitations become overwhelming.

Second, the applicability of kymographs is very limited at large. Again, the authors wipe under the table that the task of selecting profiles is left to the user. When I was accepting to review this manuscript, I was hoping to see a creative solution to this very problem. The described software is far from providing an automated solution. In general, kymographs only work in scenarios where particles repeatedly follow a stationary path. As soon as trajectories deviate from the path, e.g. because of cell movement of deformation, the trajectory lifetime is unreliable and the velocity measurement, relying on streak integration in space and time, can get unstable/impossible. And even if sufficiently stationary paths exist, e.g. because the particle motion is much faster than the confounding cell movement, the paths are in all generality curvilinear and very difficult to define. For these reasons there is a sizeable community of labs developing single particle tracking methods. I do give the authors credit, however, for picking two scenarios, where the kymograph works. Axonal particle transport is one, Paul Foscher's glued down *Aplysia* is another. In both cases it is straightforward to manually identify a general particle path, and the required time scale separation between particle dynamics and path deformation holds up. The authors should have discussed at least the rare conditions under which kymograph analysis is valid. And, in this context, it would have been advantageous to present one full example where kymograph analysis would offer a new biological insight (like most high-profile methods paper do).

Third, the authors benchmark the performance of their kymograph analysis against a particle tracker. As the senior author of the plusTipTracker paper (published 2011) used as the benchmark reference, I get the shivers when I read that 'plusTipTracker is the gold standard for microtubule tracking'. Many advances have been made by us and other labs over the past 8 years. But, this is not the issue. The objection I raise here is that a kymograph analysis is tested against a particle tracking approach (whether it is plusTipTracker or any other software) in the one scenario the kymograph approach really has advantages over particle tracking. To repeat the statement above, the kymograph works well when particles follow a stationary path. And especially with bidirectional particle motion along this path, a more generic particle tracker that does not have the prior of directional stationarity, has a higher chance for confusion. This argument is trivial. My lab, for example, has taken the kymograph approach – also in an automated fashion – when we worked on bidirectional, axonal transport (PMID: 22398725). And, in my view, one of the most impressive examples of how the particle tracking and kymograph frameworks can be algorithmically merged to increase tracking robustness in cases of path stationarity is the work by Sibarita more than a decade ago (PMID:17371444). Neither one of these papers is discussed. If the authors want to benchmark their Deep Learning streak detector, then they should use automatic kymograph analyses as a reference (there are such tools in ImageJ, etc.). And in the spirit of my first comment, they should also benchmark against approaches that use steerable line filters or curvelets to generate the same score map. I am pretty sure that the Deep Learner will be widely outperformed by a filter that is designed to detect streaks a priori.

Even if the authors were to address these points, the innovation of this manuscript does not meet the bar of what I would expect to see in a journal like *eLife*. This tool may give the authors some useful results in their own research, but I do not think this is worth more than a subsection in a Materials and methods section.

*Reviewer #3:*

In this manuscript, Jakobs et al. provide a machine-learning tool to analyze kymographs. This is a worthy addition to the state of the art, as kymograph analysis (or the related linear tracking) are notoriously difficult to perform and to automate. I think this tool could have a significant impact on a large community (researchers interested on quantifying subcellular mobility), if it can be made easily available, easy to use and broadly applicable. Related to this, I have a number of questions about the current manuscript I would like to discuss before its eventual publication in *eLife*.

The first point is about how this software is made available. Two authors of this manuscript (M. Jakobs and A. Dimitracopoulos) have launched a website/company, https://deepmirror.ai, for AI-based image processing, offering custom service and including the KymoButler software,. In addition, deepmirror.ai is providing the web-based KymoButler tool that is mentioned in the current manuscript (also, it looks like the web app now uses the deep network rather than the shallow one). This is a bit confusing, a clarification on what will be 1. open source 2. free to use (not necessarily open source) 3. commercial would be useful, as would be a more detailed "Competing Interests" section if needed.

The second point is the current limit to monodirectional tracking on kymographs. While this works for the few particular cases highlighted in the present manuscript (end-binding proteins on microtubules; actin speckles), this significantly limits the general usefulness of the tool. Kymograph analyses are generally done for bidirectional transport of cargoes (vesicles or organelles), usually along linear processes such as axons. Including in the present manuscript the extension of KymoButler to bidirectional tracking on kymographs (as is currently developed, as stated here: https://deepmirror.ai/2018/09/25/improvements-to-kymobutler/) and making it available in the open source/free tool would be a big leap in usefulness for the community. This would make a very strong case for publication in a high-profile, broad-readership journal such as *eLife*, and justify the general aspect of the current title "A deep learning software for automated kymograph anaysis".

The third point is about network robustness. AI in image analysis is very useful, but the limiting factor for its adoption by biologists is the proper validation of deep-learning tools for different images, different situations etc. Due to a lack of knowledge from potential users, an improper use of deep-learning algorithms outside of their validated range is a real concern. In the current manuscript, the authors show that their trained algorithm can ba applied to a variety of cases (EBs in different cells, actin speckles) by adjusting the prediction probability threshold p. Do the authors think that the current tool could be generally used without the need of re-training? What advice/metric would help the interested researcher to validate results on one's own data? Would it be difficult (for a non-specialist) to re-train the open-source version of KymoButler for a particular application? Is this option considered as realistic by the authors? In my experience, this is usually the bottleneck of AI approaches in image analysis, because tools can be simple to use but the training part is usually beyond the skills of a non-specialist.

[Editors’ note: what now follows is the decision letter after the authors submitted for further consideration.]

Thank you for submitting your article "KymoButler, a deep learning software for automated kymograph analysis" for consideration by *eLife*. Your article has been reviewed by three peer reviewers, and the evaluation has been overseen by a Reviewing Editor and Vivek Malhotra as the Senior Editor. The following individuals involved in review of your submission have agreed to reveal their identity: Jeff Urbach (Reviewer #1).

The reviewers have discussed the reviews with one another and the Reviewing Editor has drafted this decision to help you prepare a revised submission.

Summary:

The manuscript by Jakobs et al. describes a fully automated approach to analyze kymographs based on deep learning. The software can run on both uni- and bidirectional kymographs. The code is freely available and the tool itself is also available via a web application. This is an important state of-the-art contribution as kymographs are notoriously difficult to obtain and to automate.

Essential revisions:

We are impressed with the substantial improvements to the analysis tool and the presentation in the revised manuscript. We all agree that KymoButler is a promising tool for the field. Nevertheless, some issues remain and should be addressed.

1) The authors have shown that the current version of the software can manage unidirectional and bidirectional movements. But, the capabilities of the software should be tested more thoroughly, both on synthetic and also on "real" data. For instance:

– Synthetic data: the limit of systems performances should be explored by varying density of tracings and signal/noise ratio. Moreover, the authors should also comment on the similarities and differences of synthetic data to real kymographs.

– Real data: the tool must be tested on several examples of different real situations presented (Rab11, mitochondria, microtubule plus-ends), distinct form the training data, and average performance should be quantified against manual expert tracing and best-performing available tool.

Since the authors already have a lot of manually annotated kymographs, they could remove some of them from the training set and get hard numbers on the performance of KymoButler on such data.

In addition, for both synthetic and real data, quantitative comparison should go as far as the end result that people seek to obtain from kymographs: average directionality,% of reversals, average speed, processivity, pause duration, etc. These quantitative parameters should be compared between Kymobutler, manual expert tracing and automatic alternatives in order to evaluate how the difference in tracing detection affects the final measurements.

2) The authors should state clearly in Introduction and Discussion what can and cannot be done with their software.

Considering that biological objects move in a stochastic 3D environment, the first paragraph of the Introduction should state explicitly which classes of motions are appropriate for kymographs and for KymoButler, respectively. It is not clearly described in the current version.

What a " stationary path" means should be explained.

In addition, it would be helpful to the user to know how KymoButler manages the stochastic versus directed motions as well as the constraints on the frequency of changes in direction. These points should be added in the Discussion.

3) It also appears that KymoButler's usability can be further improved and that some issues in the program should be fixed.

For instance, when an error occurs there is no possibility to try again, but instead one has to go back to previous screen, drag the input files over again, remember the old sensitivity threshold, submit again and hope for the best. Likewise, there is no possibility to simply change the sensitivity once you see the oversegmented results. One has to go back, drag all the data over, and repeat. Considering the allowed image size, simple thresholding should be fast enough to be interactive in the browser.

From a visual inspection, it is unclear how sensitivity parameter affects the outcome. It appears that the less sensitive setting (-1) does not pick some faint traces, and also breaks trajectories in pieces, resulting in a similar trace number as at the medium sensitivity setting.

---

## [Author Response]

[Editors’ note: the author responses to the first round of peer review follow.]

This manuscript describes KymoButler, a machine-learning software that has been developed for the automated analysis of kymographs. There is a big need in this area since kymograph analysis (or related linear tracking) is notoriously difficult to perform and to automate. Although the reviewers consider that KymoButler could potentially provide an open solution to automated kymograph tracking to the community, they eventually agree after discussion that in its current state, KymoButler has not been tested on challenging enough problems and so far, does not represent a general and new solution to complicated tracking problems. In particular, the reviewers were concerned about the performance of KymoButler for tracking elements on much more diverse traces that those considered in the manuscript: for instance, in the general case of axonal mobility tracking where cargoes or components can move in a direction, change speed, immobilize, reverse direction, and do any of this several times in a single trace. We feel that in its current form, this tool does not outperform some other existing tracking methods. For these reasons, we think that your manuscript does not meet the criteria of innovation of eLife and cannot be accepted for publication in the journal.

We agree that the previously presented version of KymoButler did not apply to bidirectional cargo movement. To address this limitation, we have now developed an improved version of KymoButler, which is applicable to much more challenging tracking problems such as bidirectional cargo movement. We benchmarked the new software using complex synthetic (i.e., ground truth) data and tested it on bidirectional kymographs of a variety of biological problems including mitochondrial movement, in vitro dynamics of cytoplasmic dynein, Rab11 dynamics in axons, and microtubule plus end dynamics in dendrites. KymoButler is now able to accurately trace cargoes that move in a direction, change speed, immobilise, reverse direction, etc., even in very dense kymographs (new Figure 1 and Figure 1—figure supplement 1). It very reliably resolves crossings, junctions, and gaps, performs as well as manual data analysis by experts, and clearly outperforms current publicly available kymograph analysis tools. Our new software addresses all major concerns of the reviewers, and it will be made freely accessible to the community.

In our original submission, the presented data came from the same biological replicates as the training data. To properly challenge our software and to demonstrate that the new version of KymoButler can be applied to most if not all kymographs obtained from biological systems, we decided to replace these data in the revised manuscript with data previously published by other groups.

Reviewer #1:

This manuscript describes a very valuable contribution to the field of quantitative biological imaging. It is well written and clearly motivated, and I have no substantial concerns. The results look solid, the approach is creative and appropriate, and it seems likely that the software will be widely utilized. As with all machine learning applications, it would be nice to have a deeper understanding of how the software is actually working, but that isn't a reason not to share the results in their current form.

In the original manuscript, we have focused more on the process of extracting information from the kymographs than on how the software works. We agree that the user should have a deeper understanding of the actual workings of the software and therefore now provide a more detailed visualisation of the output of each module of the software (Figure 1—figure supplements 1 and 2).

Reviewer #2:

This manuscript describes the use of a convolutional neural net (CNN) for the analysis of kymographs. While kymography is still used by the live cell imaging community to measure subcellular dynamics, and thus some tools to automate and reduce bias would be helpful, this work does not advance the existing repertoire of image analysis for this purpose.First, the proposed method is not a kymograph analyzer, as claimed, but it relies on a CNN to learn a streak filter (the authors should have visualized the initial layers the trained net, and this would have become abundantly clear, I assume).

KymoButler does not rely on learning a streak filter, and indeed it does not use any streak filters. A streak filter is decided a priori to extract potential linear traces from an image, but does not learn the characteristic signal and background from the images. In fact, if the background has streak-like properties, the filter will classify it as signal independently of the available data. Hence, in the context of kymographs derived from live cell imaging, a streak filter is unlikely to recover many meaningful tracks, due to the excessive background fluctuation which has to be taken into account (by the user) during the design of the filter (see also our new Figure 3).

KymoButler implements a different approach. Instead of assuming that a streak filter will be the best operation to extract traces from kymographs, our software selects the best possible operations through several rounds of optimisation, which aim to best match the output of the neural network to the provided training data labels derived from manual annotation. This means that the network learns which pixels are part of a trace and which ones are not based on the available data, and not on a priori considerations. This is possible due to the incredible improvements in computation times of modern CPUs and to the adoption of GPUs that can execute an enormous number of operations in parallel. This gives us a great advantage, with the main downside being that it becomes more difficult to gain a deeper understanding of precisely how the software is working (as highlighted by reviewers #1 and #3).

To make our point clearer, we would also like to highlight the difference between the 5x5 kernels of our original neural network (Author response image 1), the 3x3 kernels of our new segmentation modules (Author response image 1), and 5x5 streak filters (Author response image 1). There are no obvious streak kernels in our approach.

**Author response image 1. respfig1:** Normalised kernel entries of our fully convolutional neural network. (**A**) Normalised 5x5 kernels from the first layer of the network from our original manuscript. (**B**) Normalised 3x3 kernels from the first layer of our unidirectional/bidirectional module. (**C**) Example 5x5 streak filters. The values run from 0 (black) to 1 (white). No obvious line filter structure is visible in our kernels.

The output of the CNN here is not a set of trajectories in the kymograph representation – the entity that eventually can be used to quantify particle velocities and, perhaps, lifetimes – but it offers merely a pixel-by-pixel map of a score that defines how probable the pixel is to be located on a streak.

This is correct and also consistent with our claims in the text. The power of CNNs is precisely this ability to identify such probability maps. However, we would like to emphasize that the probability map also takes into account the immediate neighbourhood of each pixel, to fill gaps in the images and ignore background noise.

To get from the 'streakness' score map to the individual trajectories, the authors have to threshold the map, and then apply a morphological operator to thin the high score mask to single-pixel chains. Anyone who has written a program for line or edge detection knows, these two steps are much harder and are the decisive ones for the final results.

As mentioned in the manuscript, we indeed apply morphological binarization and thinning which can be done using available codes in MATLAB or Mathematica. In our experience, this step is rather straightforward and provides excellent results as the maps are heavily de-noised and the SNR is thus very high (see Figure 1—figure supplement 1B).

These are the steps where the algorithm needs to account for streak crossings, junctions, and gaps.

The previous version of KymoButler that was included with the initial submission only analysed unidirectional particle traces, i.e. particles that did not change direction and only varied moderately in speed. As explained in the manuscript, we took care of crossings by training our neural network to only recognise traces that have negative slopes (from left upper to right lower corners). This network could then be run on the raw kymograph and its reflection to extract traces without crossings and junctions. This is possible since the data that we used to train and benchmark KymoButler does not exhibit any crossings/junctions of lines with similar slopes, i.e. they are unidirectional with almost uniform speeds. Gaps however, were taken care of by the larger convolutions of our neural network: If a particle becomes invisible for a few frames the neural network is able to bridge the gap by assigning larger scores to pixels that lie in the gap.

**Author response image 2. respfig2:** Zoom into the old Figure 1E from our manuscript. (**A**) depicts the raw kymograph and (**B**) the pixel score map from our neural network. The track highlighted in red exhibits two gaps where the particle becomes invisible for three frames each. As seen in (**B**), the network has no problem to bridge those gaps and assign high scores throughout the gap. Scale bars 2 µm (horizontal), 25 sec vertical. See also our new Figures 2 and 3, where we benchmark KymoButler (and its capability to bridge gaps) against ground truth data.

As the reviewers pointed out, the software presented in the original manuscript was unable to track bidirectional particles. However, with the new version of KymoButler this is now possible. As explained in more detail in the main text, we first use a segmentation module to filter all connected lines from a given kymograph. The resulting image comprises several thin lines that might form repeated crossings and junctions but no gaps. We then automatically trace each of these thin lines until a junction/crossing is met. To predict what happens next we designed a decision module that solves the junction/crossing by finding the most likely future path (see new Figure 1—figure supplement 1B for resulting kymographs). We are confident that this approach will allow any user to analyse their specific kymographs.

Furthermore, we now generated synthetic data with available ground truth that contained gaps and crossings/junctions, which we benchmark the performance of KymoButler against (new Figures 2 and 3). This test clearly demonstrated that KymoButler accurately identifies that vast majority of all traces, and that is does perform as well as expert human experimentalists.

The authors largely wipe this under the rug.

It was not our intention to sweep anything under the rug, and we apologize if parts of our manuscript came across this way. To avoid such misunderstandings, we expanded on the description of our approaches in the main text and added the new Figures 2 and 3 to the manuscript. Furthermore, our new KymoButler deals with these issues, and we evaluated the performance of the new software with respect to crossings, junctions, and gaps.

The results in Figures 2 and 3 reveal numerous places where exactly these two steps went wrong.

We would be grateful if the reviewer could highlight these occurrences so that we can debug our code and put more weight on training instances that show similar behavior. We could only identify 2 such cases in the kymographs shown in Figures 2/3F of our initial submission (see Author response image 3).

When we re-analyzed this figure with our new software, the two problematic points were resolved correctly (Author response image 3).

**Author response image 3. respfig3:** The kymograph shown in Figure 2 of our original manuscript analysed with the original KymoButler and the new version. We highlighted the two errors we could identify in the old version in panel B (red frames). In the upper one, a junction was not resolved properly, and in the lower one two lines were so close to each other that they were segmented as one. Neither of these errors showed up when we re-ran the data in the new KymoButler. Scale bars: 2 µm (horizontal), 25 sec (vertical).

And the authors probably intentionally pick data examples with relatively low particle density and/or fairly uniform particle motion.

We would never cherry pick data. It would be disingenuous and fraudulent, not to mention a violation of the very most basic principles of research.

The kymographs shown in the old Figures 2 and 3 were representative kymographs of our biological case study. However, we have now replaced these data with previously published, much more complex and dense kymographs, and hope that the reviewer agrees that KymoButler now performs excellently even in kymographs with high particle density and non-uniform particle motion.

As the particle density and motion heterogeneity between and within trajectories increases, these limitations become overwhelming.

We agree with the reviewer, this was a limitation of our initial software. We became aware of this problem when we were contacted by several other labs immediately after uploading our manuscript on bioRxiv, requesting a bidirectional analysis software. Hence, we have been working on a solution since, which is now included in the revised version of the manuscript.

Second, the applicability of kymographs is very limited at large. Again, the authors wipe under the table that the task of selecting profiles is left to the user.

We both agree and disagree with the sentiment of the reviewer. While kymograph applicability may seem limited, this is true of many other techniques and methods in biology (e.g. atomic force microscopy analysis / traction force microscopy analysis). However, this does not detract from how useful these methods and tools are to the researchers using them. Researchers studying neuronal transport, actin flow, or cilia/flagella, would arguably find kymographs, and new tools to analyze them reliably, extremely useful. The preprint of our software attracted significant attention (top 5% of all research outputs scored by Altmetric), and we were provided with training data by six other laboratories immediately to develop the current bidirectional KymoButler. We hope that this is a fair indicator of this work being both highly interesting and useful to many others in the community.

The task of selecting the profile along which a kymograph is made is highly specific to the biological application and is best left to a given user. Automating this process is non-trivial, particularly if we intend to make a kymograph analyser suitable for a myriad of applications. Furthermore, as mentioned in the manuscript, there are numerous packages available that make it very easy to extract kymographs from live imaging data. Hence, we did not wish to duplicate these efforts.

When I was accepting to review this manuscript, I was hoping to see a creative solution to this very problem. The described software is far from providing an automated solution. In general, kymographs only work in scenarios where particles repeatedly follow a stationary path. As soon as trajectories deviate from the path, e.g. because of cell movement of deformation, the trajectory lifetime is unreliable and the velocity measurement, relying on streak integration in space and time, can get unstable/impossible. And even if sufficiently stationary paths exist, e.g. because the particle motion is much faster than the confounding cell movement, the paths are in all generality curvilinear and very difficult to define. For these reasons there is a sizeable community of labs developing single particle tracking methods. I do give the authors credit, however, for picking two scenarios, where the kymograph works. Axonal particle transport is one, Paul Foscher's glued down Aplysia is another. In both cases it is straightforward to manually identify a general particle path, and the required time scale separation between particle dynamics and path deformation holds up. The authors should have discussed at least the rare conditions under which kymograph analysis is valid. And, in this context, it would have been advantageous to present one full example where kymograph analysis would offer a new biological insight (like most high-profile methods paper do).

We agree with the reviewer that in a large number of scenarios, kymographs cannot be applied and particle tracking is the superior method. However, in scenarios where kymographs *can* be applied, they have a number of advantages compared to particle tracking approaches, as they allow researchers to simultaneously visualise and analyse their data, and they are well suited for low signal to noise ratio data. Kymographs are applicable to many diverse studies of cytoskeletal dynamics and protein/molecular/vesicle trafficking. We feel that a profound discussion of the usefulness of kymographs is beyond the scope of the current study; however, we have now included information about when kymographs are useful in the first part of the Introduction. We would like to re-emphasize that the intent of this work is to provide a tool to the community which enables automated kymograph analysis, since many laboratories still struggle with this task.

Third, the authors benchmark the performance of their kymograph analysis against a particle tracker. As the senior author of the plusTipTracker paper (published 2011) used as the benchmark reference, I get the shivers when I read that 'plusTipTracker is the gold standard for microtubule tracking'. Many advances have been made by us and other labs over the past 8 years.

We apologise for our wording. We should have stated “plusTipTracker (and derived software packages, e.g. utrack) is a gold standard for microtubule tracking”, as it is still widely used in the community. When we started analysing microtubule plus end dynamics, we looked into available software and tested many different programs. In our hands, the plusTipTracker showed the most excellent performance. We have not only compared the performance of KymoButler against that of plusTipTracker but also against a number of other software packages but thought it only worth showing the comparison against the best performing algorithm available. We do apologize if this comparison may have come across as unflattering or critical, rather than as an advancement on what we believed to be the current best tool for the purpose.

However, we appreciate the comment of the reviewer, and as we do not focus on microtubule dynamics any longer but rather on more complex problems as suggested by the reviewers, we have replaced the comparison with the plusTipTracker by software that was designed to analyse kymographs in the revised version of the manuscript. In the new Figure 2, we benchmark KymoButler against the Fourier filtering module of Kymograph Direct, and in the new Figure 3 against a custom-written wavelet filtering algorithm as suggested by the reviewer (see below), as Kymograph Direct here did not return any meaningful data (Figure 3—figure supplement 1).

But, this is not the issue. The objection I raise here is that a kymograph analysis is tested against a particle tracking approach (whether it is plusTipTracker or any other software) in the one scenario the kymograph approach really has advantages over particle tracking. To repeat the statement above, the kymograph works well when particles follow a stationary path. And especially with bidirectional particle motion along this path, a more generic particle tracker that does not have the prior of directional stationarity, has a higher chance for confusion. This argument is trivial. My lab, for example, has taken the kymograph approach – also in an automated fashion – when we worked on bidirectional, axonal transport (PMID: 22398725).

As mentioned above, we agree with the reviewer that there are many scenarios where kymographs cannot be applied, and particle tracking approaches are superior, and we apologize for not explicitly stating this in the original version of our manuscript. In the revised Introduction, we now acknowledge the importance of particle tracking approaches and explain in which specific scenarios kymographs are useful.

We thank the reviewer for the reference provided, which we now cite. However, we are unsure to what extent this publication is comparable to our manuscript. In the Materials and methods section of that paper, the following statement is made: “The output of this step contains mostly trajectory segments, so an additional process was performed to link these segments into full trajectories using a multiple hypothesis-testing algorithm (Blackman, 1999). A manual process was subsequently used to recover trajectories the software was unable to recover. Finally, all recovered trajectories were *individually inspected, and errors were corrected*.” We are unclear on how reliably the automated parts of this process work without manual supervision, and as no open source software was provided with the paper, we could not compare it with our tool easily. If the software could be made publically available, we would be happy to try it.

And, in my view, one of the most impressive examples of how the particle tracking and kymograph frameworks can be algorithmically merged to increase tracking robustness in cases of path stationarity is the work by Sibarita more than a decade ago (PMID:17371444). Neither one of these papers is discussed.

We are very grateful for bringing this publication to our attention, which we now also cite. The authors here very nicely demonstrated the power and versatility of 4D kymogram analysis. We hope that we can also extend our approach to 4D in the future. However, while the authors of PMID:17371444 developed an automated approach for the extraction of tracks from 4D kymograms, the software does not appear to be publically available. Additionally, the analysed cells were *Hela* cells that overexpressed pEGFP-Rab6, leading to very high signal to noise ratios (but potentially also to changes in protein dynamics) in contrast to endogenously expressed EB1GFP encountered in many studies on microtubule dynamics and in our original manuscript. Hence, it is not clear how versatile this software is or how applicable it is to the type of questions we (and other groups using kymograph analysis) ask.

If the authors want to benchmark their Deep Learning streak detector, then they should use automatic kymograph analyses as a reference (there are such tools in ImageJ, etc.). And in the spirit of my first comment, they should also benchmark against approaches that use steerable line filters or curvelets to generate the same score map. I am pretty sure that the Deep Learner will be widely outperformed by a filter that is designed to detect streaks a priori.

We would like to reiterate that KymoButler is not a streak detector (see Author response image 2). We are not aware of any fully automated kymograph analysis tools in ImageJ, plugins such as MultiKymograph, KymoGraphBuilder and other tools, e.g. KymographTracker (ICY), can be used to generate and threshold kymographs but require manual ROI selection for data analysis. We would be grateful if the reviewer could advise us on existing algorithms that are fully automated.

As suggested by the reviewer, we now benchmark the unidirectional module of KymoButler against KymographDirect (PMID: 27099372), which implements semi-automated analysis through Fourier filtering and local thresholding. Additionally, we benchmark the bidirectional module against a custom wavelet filtering algorithm as we did not obtain any satisfactory results with any available software we tested (Figure 3—figure supplement 1). Both tests showed that our Deep Learning Software clearly outperforms any other publically available software; it actually performs as accurately as manual expert annotation, just much faster and without any unconscious bias.

Even if the authors were to address these points, the innovation of this manuscript does not meet the bar of what I would expect to see in a journal like eLife. This tool may give the authors some useful results in their own research, but I do not think this is worth more than a subsection in a Materials and methods section.

The cloud-based software has been used on over 1800 unique images by the wider community since the paper was uploaded on bioRxiv 5 months ago. We have received plenty of positive feedback on the use of KymoButler for kymograph analysis workflows from many other groups at different institutions worldwide at conferences, via email, and even through social media. Hence, we strongly believe that KymoButler is very useful to a much broader base than our research group alone. We are now in close collaboration with several of these laboratories, who are sending us their data, and helping us improve our network further. The inclusion of bidirectional tracking in our fully automated KymoButler is certainly of wide interest to an even larger community already working with kymographs. Furthermore, a fully automated software has great potential to convince more groups to work with kymographs in the future.

Reviewer #3:

In this manuscript, Jakobs et al. provide a machine-learning tool to analyze kymographs. This is a worthy addition to the state of the art, as kymograph analysis (or the related linear tracking) are notoriously difficult to perform and to automate. I think this tool could have a significant impact on a large community (researchers interested on quantifying subcellular mobility), if it can be made easily available, easy to use and broadly applicable. Related to this, I have a number of questions about the current manuscript I would like to discuss before its eventual publication in eLife.

We would like to thank the reviewer for this very positive evaluation of the impact of our tool.

The first point is about how this software is made available. Two authors of this manuscript (M. Jakobs and A. Dimitracopoulos) have launched a website/company, https://deepmirror.ai, for AI-based image processing, offering custom service and including the KymoButler software,. In addition, deepmirror.ai is providing the web-based KymoButler tool that is mentioned in the current manuscript (also, it looks like the web app now uses the deep network rather than the shallow one). This is a bit confusing, a clarification on what will be 1. open source 2. free to use (not necessarily open source) 3. commercial would be useful, as would be a more detailed "Competing Interests" section if needed.

We apologise for the confusion the webpage might have caused. We launched deepmirror.ai as a platform to promote the use of AI-based technologies for biological data analysis. We will be publishing tutorials and sample code to help people get started with developing their own machine learning software. We also intend to publish our work on KymoButler and future publications of our AI-based software on the website. All of this will be free of charge and available to all. Further in the future, we plan to also start offering paid professional services for customers who want to set up custom AI-based software for applications, in case they are not covered by our research. This software may or may not be made available on deepmirror.ai, depending on our clients’ requests. We amended the “Competing Interests” section in our manuscript to clarify our position.

We apologise for changing the neural network in the webform during the review process. We realised that some of our users had much better results with the deeper version, and hence decided to optimise that network and make it available on the Cloud. We also include a link to a Mathematica library in the manuscript, as a set of functions that can be easily integrated into other workflows. Additionally, we provide a link to a Mathematica script which enables experienced users to train their own network. We would like to reiterate that the software presented in this paper will not be commercialized, but will be freely available to every user.

The second point is the current limit to monodirectional tracking on kymographs. While this works for the few particular cases highlighted in the present manuscript (end-binding proteins on microtubules; actin speckles), this significantly limits the general usefulness of the tool. Kymograph analyses are generally done for bidirectional transport of cargoes (vesicles or organelles), usually along linear processes such as axons. Including in the present manuscript the extension of KymoButler to bidirectional tracking on kymographs (as is currently developed, as stated here: https://deepmirror.ai/2018/09/25/improvements-to-kymobutler/) and making it available in the open source/free tool would be a big leap in usefulness for the community. This would make a very strong case for publication in a high-profile, broad-readership journal such as eLife, and justify the general aspect of the current title "A deep learning software for automated kymograph anaysis".

We thank the reviewer for this excellent suggestion and the encouraging comments on the importance and usefulness of our software. As mentioned in the general comments above, the new version of KymoButler allows exactly what the reviewer recognizes to be needed by the community. The improved KymoButler will replace our old version and be made available free-to-use.

The third point is about network robustness. AI in image analysis is very useful, but the limiting factor for its adoption by biologists is the proper validation of deep-learning tools for different images, different situations etc. Due to a lack of knowledge from potential users, an improper use of deep-learning algorithms outside of their validated range is a real concern. In the current manuscript, the authors show that their trained algorithm can ba applied to a variety of cases (EBs in different cells, actin speckles) by adjusting the prediction probability threshold p. Do the authors think that the current tool could be generally used without the need of re-training?

We are glad that the reviewer touched upon this important point, which is true of most (if not all) automated data analysis tools. Often, a network trained for a specific task can result in catastrophic artefacts when applied to a different problem. However, the new bidirectional KymoButler should be applicable to most if not all Kymograph problems. Nevertheless, it is good practice for users to visually inspect analysed kymographs to confirm the suitability of this tool. In our web-based application, the input image is overlaid with KymoButler’s predicted traces before the result is downloaded to ensure that no result is obtained without a visual inspection. In addition, we have a webform should users need to contact us with any concerns, e.g. if the network produces unexpected results. We now explicitly point at this in the Discussion.

What advice/metric would help the interested researcher to validate results on one's own data?

We would strongly suggest comparing any automated software to manual annotation as the best benchmark, which we now mention in the Discussion.

Would it be difficult (for a non-specialist) to re-train the open-source version of KymoButler for a particular application? Is this option considered as realistic by the authors? In my experience, this is usually the bottleneck of AI approaches in image analysis, because tools can be simple to use but the training part is usually beyond the skills of a non-specialist.

The bidirectional KymoButler should be applicable to all kymographs from biological data, and the software will constantly be updated as users upload their kymographs. However, should the user wish to repurpose and retrain our software, the code is freely available (see above). This could be a bottleneck, as it is not straightforward to retrain the networks with new data. Nonetheless, it should be possible for a non-specialist with some experience in programming to accomplish, and we will design tutorials for this purpose and make them available on our website. In addition, we will be contactable for consultations, should a user face difficulties with repurposing the networks.

[Editors’ note: the author responses to the re-review follow.]

Essential revisions:We are impressed with the substantial improvements to the analysis tool and the presentation in the revised manuscript. We all agree that KymoButler is a promising tool for the field. Nevertheless, some issues remain and should be addressed.1) The authors have shown that the current version of the software can manage unidirectional and bidirectional movements. But, the capabilities of the software should be tested more thoroughly, both on synthetic and also on "real" data. For instance:– Synthetic data: the limit of systems performances should be explored by varying density of tracings and signal/noise ratio. Moreover, the authors should also comment on the similarities and differences of synthetic data to real kymographs.

This is an excellent suggestion. We added a new supplementary figure (Figure 1—figure supplement 5) investigating the effects of increasing noise and density on the track recall/precision for both unidirectional and bidirectional kymographs. We find that KymoButler performs well down to SNRs of ~1.2 and a signal coverage of up to ~60%, similar to manual analysis at comparable SNRs and signal coverage. We also now discuss the differences between synthetic and real data in the Discussion (paragraph five).

– Real data: the tool must be tested on several examples of different real situations presented (Rab11, mitochondria, microtubule plus-ends), distinct form the training data, and average performance should be quantified against manual expert tracing and best-performing available tool.Since the authors already have a lot of manually annotated kymographs, they could remove some of them from the training set and get hard numbers on the performance of KymoButler on such data.In addition, for both synthetic and real data, quantitative comparison should go as far as the end result that people seek to obtain from kymographs: average directionality,% of reversals, average speed, processivity, pause duration, etc. These quantitative parameters should be compared between Kymobutler, manual expert tracing and automatic alternatives in order to evaluate how the difference in tracing detection affects the final measurements.

As previously mentioned in the Materials and methods, we already had both a training and a validation data set. We could not disclose our raw training and validation data in this manuscript, as all of the data are currently unpublished, and we agreed with all laboratories providing us with their (unpublished) kymographs not to publish the raw data. However, to address this important point, we now added two supplementary figures to the manuscript (Figure 2—figure supplement 1, Figure 3—figure supplement 2) that quantify KymoButler’s relative average performance against manual expert tracing and best-performing available tool on synthetic as well as real data, investigating all parameters mentioned above.

The real data used in our new comparison include 6 kymographs with labelled microtubule plus ends for the unidirectional analysis, 2 kymographs showing axonal mitochondria dynamics, one showing molecular motor processivity, and 3 kymographs depicting vesicle movement in axons and dendrites. Since we do not have a ground truth for the real kymographs, we cannot compare recall/precision, gap detection, and crossing detection in those.

2) The authors should state clearly in Introduction and Discussion what can and cannot be done with their software.Considering that biological objects move in a stochastic 3D environment, the first paragraph of the Introduction should state explicitly which classes of motions are appropriate for kymographs and for KymoButler, respectively. It is not clearly described in the current version.What a " stationary path" means should be explained.

We are grateful for these suggestions and changed the first paragraph of the Introduction to clarify when kymographs and hence KymoButler should be used. Additionally, we now explain what a stationary path is in both the Introduction and the legend of Figure 1A.

In addition, it would be helpful to the user to know how KymoButler manages the stochastic versus directed motions as well as the constraints on the frequency of changes in direction. These points should be added in the Discussion.

We added more quantification of KymoButler’s performance on both directed (Figure 2—figure supplement 1) and stochastic motion (Figure 3—figure supplement 2) including a quantification of the likelihood of reversals. Notably, KymoButler does not have any restraints on the frequency of changes in direction (stochasticity) as shown in Author response image 4. However, if an increase in stochastic movements leads to an increase in the number of crossings in dense kymographs, KymoButler’s performance is affected adversely (see new Figure 1—figure supplement 5), which however would also be true for expert manual data analysis. We now discuss this in paragraph two of the Discussion.

**Author response image 4. respfig4:** Track recall and precision as a function of number of particle direction changes.

3) It also appears that KymoButler's usability can be further improved and that some issues in the program should be fixed.For instance, when an error occurs there is no possibility to try again, but instead one has to go back to previous screen, drag the input files over again, remember the old sensitivity threshold, submit again and hope for the best. Likewise, there is no possibility to simply change the sensitivity once you see the oversegmented results. One has to go back, drag all the data over, and repeat. Considering the allowed image size, simple thresholding should be fast enough to be interactive in the browser.From a visual inspection, it is unclear how sensitivity parameter affects the outcome. It appears that the less sensitive setting (-1) does not pick some faint traces, and also breaks trajectories in pieces, resulting in a similar trace number as at the medium sensitivity setting.

We are very grateful for the reviewer’s suggestions how to improve our software. We changed the webform so that one first uploads a kymograph now and then dynamically adjusts the threshold, allowing fast screening of different segmentations. Note that simple thresholding only applies to unidirectional kymographs, while bidirectional track segmentation takes longer as here paths have to be traced. Additionally, we now also provide an application programming interface (upon request), that enables programmers to call our software from any other programming language to incorporate it in their workflow. We also removed the misleading sensitivity parameter and replaced it with a single threshold value, which is the same as the threshold discussed in the manuscript. Finally, we added more post-processing functionality by allowing users to discard small fragmented tracks and added quantity averages and plots to the output.